# Transition Constrained Bayesian Optimization via Markov Decision Processes

**Jose Pablo Folch**
Imperial College London
London, UK

**Calvin Tsay**
Imperial College London
London, UK

**Robert M Lee**
BASF SE
Ludwigshen, Germany

**Behrang Shafei**
BASF SE
Ludwigshen, Germany

**Weronika Ormaniec**
ETH Zurich
Zurich, Switzerland

**Andreas Krause**
ETH Zurich
Zurich, Switzerland

**Mark van der Wilk**
Imperial College London
London, UK

**Ruth Misener**
Imperial College London
London, UK

**Mojmír Mutný**
ETH Zurich
Zurich, Switzerland

## Abstract

Bayesian optimization is a methodology to optimize black-box functions. Traditionally, it focuses on the setting where you can arbitrarily query the search space. However, many real-life problems do not offer this flexibility; in particular, the search space of the next query may depend on previous ones. Example challenges arise in the physical sciences in the form of local movement constraints, required monotonicity in certain variables, and transitions influencing the accuracy of measurements. Altogether, such *transition constraints* necessitate a form of planning. This work extends classical Bayesian optimization via the framework of Markov Decision Processes. We iteratively solve a tractable linearization of our utility function using reinforcement learning to obtain a policy that plans ahead for the entire horizon. This is a parallel to the optimization of an *acquisition function in policy space*. The resulting policy is potentially history-dependent and non-Markovian. We showcase applications in chemical reactor optimization, informative path planning, machine calibration, and other synthetic examples.

## 1 Introduction

Many areas in the natural sciences and engineering deal with optimizing expensive black-box functions. Bayesian optimization (BayesOpt) [1–3], a method to optimize these problems using a probabilistic surrogate, has been successfully applied to a myriad of examples, e.g. hyper-parameter selection [4], robotics [5], battery design [6], laboratory equipment tuning [7], and drug discovery [8]. However, state-of-the-art algorithms are often ill-suited when physical sciences interact with potentially dynamic systems [9]. In such circumstances, real-life constraints limit our future decisions while depending on the prior state of our interaction with the system. This work focuses on transition constraints influencing future choices depending on the current state of the experiment. In other words, reaching certain parts of the decision space (search space) requires long-term planning in our optimization campaign. This effectively means we address a general sequential-decision problem akin to those studied in reinforcement learning (RL) or optimal control for the task of optimization. We assume the transition constraints are known *a priori* to the optimizer.

Applications with transition constraints include chemical reaction optimization [10–12], environmental monitoring [13–17], lake surveillance with drones [18–20], energy systems [21], vapor compression systems [22], electron-laser tuning [23] and seabed identification [24]. For example,

Figure 1 depicts an application in environmental monitoring where autonomous sensing vehicles must avoid obstacles (similar to Hitz et al. [18]). Our main focus application are transient flow reactors [25–27]. Such reactors allow efficient data collection by obtaining semi-continuous time-series data rather than a single measurement after reaching the steady state of the reactor. As we can only change the inputs of the reactor continuously and slowly to maintain quasi-steady-state operation, allowing arbitrary changes, as in conventional BayesOpt, would result in measurement sequences which are not possible due to physical limitations.

**Problem Statement.** More formally, we design an algorithm to identify the optimal configuration of a physical system governed by a black box function $f$, namely, $x^\star = \arg\max_{x \in \mathcal{X}} f(x)$. The set $\mathcal{X}$ summarizes all possible system configurations, the so called *search space*. We assume that we can sequentially evaluate the unknown function at specific points $x$ in the search space and obtain noisy observations, $y = f(x) + \epsilon(x)$, where $\epsilon$ has a known Gaussian likelihood, which is possibly heteroscedastic. We assume that $f$ can be modeled probabilistically using a Gaussian process prior that we introduce later. Importantly, the order of the evaluations is dictated by *known*, potentially stochastic, dynamics modeled by a Markov chain that limits our choices of $x \in \mathcal{X}$.

**BayesOpt with a Markov Decision Processes.** The problem of maximizing an unknown function could be addressed by BayesOpt, which typically chooses to query $f(x)$ by sequentially maximizing an *acquisition function*, $u$:

$$x_{t+1} = \arg\max_{x \in \mathcal{X}} u(x|\mathbf{X}_t), \tag{1}$$

depending on all the past data at iteration $t$, $\mathbf{X}_t$. Eq. (1) arises as a greedy one-step approximation whose overall goal is to minimize e.g. cumulative regret, and assumes that any choice of point in the search space $\mathcal{X}$ is available. However, given transition constraints, we must traverse the search space according to the system dynamics. This work extends the BayesOpt framework and provides a method that constructs a potentially non-Markovian policy by myopically optimizing a utility as,

$$\pi_{t+1} = \arg\max_{\pi \in \Pi} \mathcal{U}(\pi|\mathbf{X}_t), \tag{2}$$

where $\mathcal{U}$ is the greedy utility of the policy $\pi$ and $\mathbf{X}_t$ encodes past trajectories through the search space. In the following sections, we will show how to tractably formulate the overall utility, how to greedily maximize it, and how to adapt it to admit policies depending on the full optimization history.

**Contributions.** We present a BayesOpt framework that tractably plans over the complete experimentation horizon and respects Markov transition constraints, building on active exploration in Markov chains [17]. Our key contributions include:

- We identify a novel utility function for maximum identification as a function of policies, and greedily optimize it. The optimization is tractable, and does not scale exponentially in the policy horizon. In many cases, the problem is convex in the natural representation.

- We provide exact solutions to the optimization problems using convex optimization for discrete Markov chains. For continuous Markov chains, we propose a reparameterization by viewing our problem as an instance of model predictive control (MPC) with a non-convex objective. Interestingly, in both cases, the resulting policies are history-dependent (non-Markovian).

- We analyze the scheme theoretically and empirically demonstrate its practicality on problems in physical systems, such as electron laser calibration and chemical reactor optimization.

## 2 Background

Because our contributions address experimental design of real-life systems by intersecting design of experiments, BayesOpt and RL, we review each of these of components. Refer to Figure 5 in Appendix A for a visual overview of how we selected the individual components for tractability of the entire problem.

**Gaussian Processes** To model the unknown function $f$, we use Gaussian processes (GPs) [28]. GPs are probabilistic models that capture nonlinear relationships and offer well-calibrated uncertainty estimates. Any finite marginal of a GP, e.g., for inputs $(x_1, .., x_p)$, the values $\{f(x_j)\}_{j=1}^p$, are normally distributed. We adopt a Bayesian approach and assume $f$ is a sample from a GP prior with a known covariance kernel, $k$, and zero mean function, $f \sim \mathrm{GP}(0, k)$. Under these assumptions, the posterior of $f$, given a Gaussian likelihood of data, is a GP that is analytically tractable.

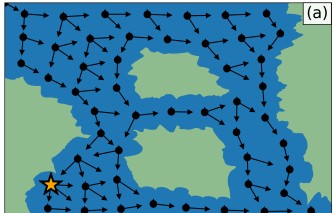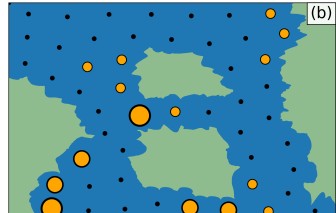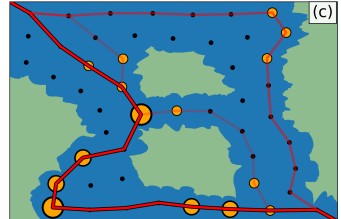

Figure 1: Representative task of finding pollution in a river while following the current. (a) Problem formulation: The star represents the maximizer and the arrows the Markov dynamics. (b) Objective formulation: Orange balls represent potential maximizers, with size corresponding to model uncertainty. (c) Optimization: Deploy a potentially stochastic policy that minimizes our objective.

## 2.1 Maximum Identification: Experiment Design Goal

Classical BayesOpt is naturally myopic in its definition as a greedy one-step update (see (1)), but has the overall goal to minimize, e.g., the cumulative regret. Therefore $u$ needs to chosen such that overall non-myopic goals can be achieved, usually defined as balancing an exploration-exploitation trade-off. In this paper we follow similar ideas; however, we do not focus on regret but instead on gathering information to maximize our chances to identify $x^\star$, the maximizer of $f$.

**Maximum Identification via Hypothesis testing.** Maximum identification can naturally be expressed as a multiple hypothesis testing problem, where we need to determine which of the elements in $\mathcal{X}$ is the maximizer. To do so, we require good estimates of the differences (or at least their signs) between individual queries $f(x_i) - f(x_j)$; $x_i, x_j \in \mathcal{X}$. For example, if $f(x_i) - f(x_j) \leq 0$, then $x_i$ cannot be a maximizer. Given the current evidence, the set of arms which we cannot rule out are all *potential maximizers*, $\mathcal{Z} \subset \mathcal{X}$. At termination we report our best guess for the maximizer as:

$$x_T = \arg\max_{x \in \mathcal{Z}} \mu_T(x), \qquad \text{where } \mu_T \text{ is the predictive mean at termination time } T.$$

Suppose we are in step $t$ out of $T$, then let $\mathbf{X}_t$ be the set of previous queries, we seek to identify new $\mathbf{X}_{\text{new}}$ that when evaluated minimize the probability of returning a sub-optimal arm at the end. For a given function draw $f$, the probability of returning a wrong maximizer $z \neq x_f^\star$ is $P(\mu_T(z) - \mu_T(x_f^\star) \geq 0|f)$. We can then consider the *worst-case* probability across potential maximizers, and taking expectation over $f$ we obtain a utility through an asymptotic upper-bound on the log-probability, indeed for large $T$ we obtain:

$$\mathbb{E}_f \left[ \sup_{z \in \mathcal{Z} \setminus \{x_f^\star\}} \log P(\mu_T(z) - \mu_T(x_f^\star) \geq 0|f) \right] \overset{.}{\leq} -\frac{1}{2} \mathbb{E}_f \left[ \sup_{z \in \mathcal{Z} \setminus \{x_f^\star\}} \frac{(f(z) - f(x_f^\star))^2}{k_{\mathbf{X}_t \cup \mathbf{X}_{new}}(z, x_f^\star)} \right] \quad (3)$$

The expectation is on the current prior (posterior up to $\mathbf{X}_t$), the kernel $k$ is the posterior kernel given observations $\mathbf{X}_t \cup \mathbf{X}_{\text{new}}$. Since we consider the probability of an error, it is more appropriate to talk about minimizing instead of 'maximizing the utility' but the treatment is analogous. Further, note the intuitive interpretation of the bound: the probability of an error will be minimized if the uncertainty is small or if the values of $f(z)$ and $f(x_f^\star)$ are far apart. The non-trivial distribution of $f(x^\star)$ [29] renders the utility intractable; therefore we employ a simple and tractable upper bound on the objective (3) which can be optimized by minimizing the uncertainty among all pairs in $\mathcal{Z}$:

$$U(\mathbf{X}_{\text{new}}) = \max_{z', z \in \mathcal{Z}, z \neq z'} \text{Var}[f(z) - f(z')|\mathbf{X}_t \cup \mathbf{X}_{\text{new}}]. \quad (4)$$

Such objectives can be solved greedily in a similar way as acquisition functions in Eq. (1) by minimizing $U$ over $\mathbf{X}_{\text{new}}$. Note that Fiez et al. [30] derive this objective for the same problem with linear bandits, albeit they consider the frequentist setting and (surprisingly) a different optimality criterion: minimizing $T$ for a given failure rate. For their setting, the authors prove that it is an asymptotically optimal objective to follow. They do not consider any Markov chain structure. Derivation of the Bayesian utility and its upper bound in Eq.(4) can be found in Appendix C.1–C.2.

**Utility with kernel embeddings.** For illustrative purposes, consider a special case where the kernel $k$ has a low rank due to existence of embeddings $\Phi(x) \in \mathbb{R}^m$, i.e., $k(x, y) = \Phi(x)^\top \Phi(y)$. Such embeddings can be, e.g., Nyström features [31] or Fourier features [32, 33]. While not necessary,

these formulations make the objectives considered in this work more tractable and easier to expose to the reader. With the finite rank assumption, the random function $f$ becomes,

$$f(x) = \Phi(x)^T\theta \quad \text{and} \quad \theta \sim \mathcal{N}(0, \mathbf{I}_{m \times m}) \tag{5}$$

where $\theta$ are weights with a Gaussian prior. We can then rewrite the objective Eq. (4) as:

$$U(\mathbf{X}_{\text{new}}) = \max_{z,z' \in \mathcal{Z}} ||\Phi(z) - \Phi(z')||^2_{\left(\sum_{x \in \mathbf{x}_t \cup \mathbf{x}_{\text{new}}} \frac{\Phi(x)\Phi(x)^\top}{\sigma^2} + \mathbf{I}\right)^{-1}}. \tag{6}$$

This reveals an essential observation that the utility depends only on the visited states; not their order. This suggests a vast simplification, where we do not to model whole trajectories, and Markov decision processes sufficiently describe our problem. Additionally, numerically, the objective involves the inversion of an $m \times m$ matrix instead of $|\mathcal{X}| \times |\mathcal{X}|$ (see Sec. 4). Appendix D.1 provides a utility without the finite rank-assumptions that is more involved symbolically and computationally.

## 2.2 Markov Decision Processes

To model the transition constraints, we use the versatile model of Markov Decision processes (MDPs). We assume an environment with state space $\mathcal{X}$ and action space $\mathcal{A}$, where we interact with an unknown function $f : \mathcal{X} \times \mathcal{A} \to \mathbb{R}$ by rolling out a policy for $H$ time-steps (horizon) and obtain a trajectory, $\tau = (x_0, a_0, x_1, a_1, ..., x_{H-1}, a_{H-1})$. From the trajectory, we obtain a sequence of noisy observations $y(\tau) := \{y(x_0, a_0), ..., y(x_{H-1}, a_{H-1})\}$ s.t. $y(x_h) = f(x_h, a_h) + \epsilon(x_h, a_h)$, where $\epsilon(x_h, a_h)$ is zero-mean Gaussian with known variance which is potentially state and action dependent. The trajectory is generated using a *known* transition operator $P(x_{h+1}|x_h, a_h)$. A Markov policy $\pi(a_h|x_h)$ is a mapping that dictates the probability of action $a_h$ in state $x_h$. Hence, the state-to-state transitions are $P(x_{h+1}, x_h) = \sum_{a \in \mathcal{A}} \pi_h(a|x_h)P(x_{h+1}|x_h, a)$. In fact, an equivalent description of any Markov policy $\pi$ is the corresponding distribution giving us the probability of visiting a state-action pair under the policy, which we denote $d_\pi \in \mathcal{D}$, where

$$\mathcal{D} := \left\{ \forall h \in [H]\, d_h \mid d_h(x, a) \geq 0, \ \sum_{a,x} d_h(x, a) = 1, \ \sum_a d_h(x', a) = \sum_{x,a} d_{h-1}(x, a)p(x'|x, a) \right\}$$

We will use this polytope to reformulate our optimization problem over trajectories. Any $d \in \mathcal{D}$ can be realized by a Markov policy $\pi$ and vice-versa. We work with non-stationary policies, meaning the policies depend on horizon count $h$. The execution of deterministic trajectories is only possible for deterministic transitions. Otherwise, the resulting trajectories are random. In our setup, we repeat interactions $T$ times (episodes) to obtain the final dataset of the form $\mathbf{X}_T = \{\tau_i\}_{i=1}^T$.

## 2.3 Experiment Design in Markov Chains

Notice that the utility $U$ in Eq. 6 depends on the states visited and hence states of the trajectory. In our notation, $\mathbf{X}_t$ will now form a set of executed trajectories. With deterministic dynamics, we could optimize over trajectories, but this would lead to an exponential blowup (i.e. $|X|^H$). In fact, for stochastic transitions, we cannot pick the trajectories directly, so instead we work in the space of distributions. For a given policy, through sampling, we are able to create an empirical distribution of all the state-action pairs visited during policy executions, $\hat{d}_\pi(x, a)$, which assigns equal mass to each state-action visited during our trajectories. This allows us to focus on the expected utility over the randomness of the policy and the environment, namely,

$$\mathcal{U}(d_\pi) := U(\mathbb{E}_{\tau_1 \sim \pi_1, ...\tau_t \sim \pi_t}[\hat{d}_\pi]). \tag{7}$$

This formulation stems from Mutný et al. [17] who try to tractably solve such objectives that arise in experiment design by performing planning in MDPs. They focus on learning linear operators of an unknown function, unlike identifying a maximum, as we do here. The key observation they make is that any policy $\pi$ induces a distribution over the state-action visitations, $d_\pi$. Therefore we can reformulate the problem of finding the optimal policy, into finding the optimal distribution over state-action visitations as: $\min_{d_\pi \in \mathcal{D}} \mathcal{U}(d_\pi)$, and then construct policy $\pi$ via marginalization. We refer to this optimization as the *planning problem*. The constraint $\mathcal{D}$ encodes the dynamics of the MDP.

## 2.4 Additional Related Works

The most relevant prior work to ours is exploration in reinforcement learning through the use of Markov decision processes as in Mutný et al. [17] and convex reinforcement learning of Hazan et al. [34], Zahavy et al. [35] which we will use to optimize the objective. Other related works are:

**Pure exploration bandits objectives.** Similar objectives to ours have been explored for BayesOpt. Li and Scarlett [36] use the $\mathcal{G}$-allocation variant of our objective for batch BayesOpt, achieving good theoretical bounds. Zhang et al. [37] and recently Han et al. [38] take advantage of possible maximizer sets to train localized models, while Salgia et al. [39] show that considering adaptive maximization sets yields good regret bounds under random sampling. Contrary to them, motivation and derivation in terms of a Bayesian decision rule do not appear elsewhere according to our best knowledge. We also recognize that we can relax the objective and optimize it in the space of policies.

**Optimizing over sequences.** Previous work has focused on planning experimental sequences for minimizing switching costs [11, 21, 40, 41] however they are only able adhere to strict constraints under truncation heuristics [20, 22, 42]. Recently, Qing et al. [43] also tackle Bayesian optimization within dynamical systems, with the focus of optimizing initial conditions. Concurrent work of Che et al. [44] tackles a constrained variant of a similar problem using model predictive control with a different goal.

**Regret vs Best-arm identification.** Most algorithms in BayesOpt focus on regret minimization. This work focuses on maximizer identification directly, i.e., to identify the maximum after a certain number of iterations with the highest confidence. This branch of BayesOpt is mostly addressed in the bandit literature [45]. Our work builds upon prior works of Soare et al. [46], Yu et al. [47], and specifically upon the seminal approach of Fiez et al. [30] to design an optimal objective via hypothesis testing. Novel to our setting is the added difficulty of transition constraints necessitating planning.

**Non-myopic Bayesian Optimization.** Look-ahead BayesOpt [48–54] seeks to improve the greedy aspect of BayesOpt. Such works also use an MDP problem formulation, however, they define the state space to include all past observations (e.g. [55, 56]). This comes at the cost of simulating expensive integrals, and the complexity grows exponentially with the number of look-ahead steps (usually less than three steps). Our work follows a different path, we maintain the greedy approach to control computational efficiency (i.e. by optimizing over the space of Markovian policies), and maintain provable and state-of-art performance. Even though the optimal policy through non-myopic analysis is non-Markovian, in Sec. 4, we show that *adaptive resampling* iteratively approximates this non-myoptic optimal policies in a numerically tractable way via receeding horizon planning. In our experiments we comfortably plan for over a hundred steps.

## 3 Transition Constrained BayesOpt

This section introduces BayesOpt with transition constraints. We use MDPs to encode constraints. Namely, the Markov dynamics dictates which inputs we are allowed to query at time-step $h+1$ given we previously queried state $x_h$. This mean that the transition operator is $P(x_{h+1}|x_h, a) = 0$ for any transition $x_h \to x_{h+1}$ not allowed by the physical constraints.

Motivated by our practical experiments with chemical reactors, we distinguish two different types of *feedback*. With **episodic feedback** we can be split the optimization into episodes. At the end of each episode of length $H$, we obtain the whole set of noisy observations. On the other hand, **instant feedback** is the setting where we obtain a noisy observation immediately after querying the function. *Asynchronous feedback* describes a mix of the previous two, where we obtain observations with unspecific a delay.

### 3.1 Expected Utility for Maximizer Identification

In section 2.1 we introduced the utility for maximum identification. Using the same simplifying assumption (finite rank approximation of GPs in Sec. 2.1, Eq. (4)), we can show that the expected utility $\mathcal{U}$ can be rewritten in terms of the state-action distribution induced by $\mathbf{X}_{\text{new}}$:

$$\mathcal{U}(d_\pi) = \max_{z,z' \in \mathcal{Z}} ||\Phi(z) - \Phi(z')||^2_{\mathbf{V}(d_\pi)^{-1}} \tag{8}$$

where $\mathbf{V}(d_\pi) = \left( \sum_{x,a \in \mathcal{X} \times \mathcal{A}} \frac{d_\pi(x,a)\Phi(x,a)\Phi(x,a)^\top}{\sigma^2(x,a)} + \frac{1}{TH}\mathbf{I} \right)$. The variable $d_\pi(x,a)$ is a state-action visitation, $\Phi(x)$ are e.g. Nyström features of the GP. We prove that the function is additive in terms of state-action pairs in Lemma D.1 in Appendix D, a condition required for the expression as a function of state-action visitations [17]. Additionally, by rewriting the objective in this form, the dependence and convexity with respect to the state-action density $d_\pi$ becomes clear as it is only composition of a linear function with an inverse operator. Also, notice that the constraint set is a convex polytope. Therefore we are able to use convex optimization to solve the planning problem (see Sec. 4).

---

**Algorithm 1** Transition Constrained BayesOpt via MDPs

---

**Input:** Procedure for estimating sets of maximizers, initial point $x_0$, initial set of maximizer candidates $\mathcal{Z}_0$
Initialize the empirical state-action distribution $\hat{d}_0 = 0$
**for** $t = 0$ **to** $T - 1$ **do**
  **for** $h = 0$ **to** $H - 1$ **do**
    $\mathcal{U}_{t,h}(d_\pi) \leftarrow \mathcal{U}(d_\pi \oplus \hat{d}_{t,h} | \mathcal{Z}_{t,h}, x_{t,h})$         // *define the objective, see eq.* (8)
    $\pi_{t,h} = \arg\min_{\pi : d_\pi \in \mathcal{D}_{t,h}} \mathcal{U}_{t,h}(d_\pi)$         // *solve MDP planning problem*
    $x_{t,h+1} = \pi_{t,h}(x_{t,h})$         // *deploy policy*
    **if** feedback is immediate **then**
      $y_{t,h+1} = f(x_{t,h+1}) + \epsilon_{t,h}$         // *obtain observation*
      $\mathcal{GP}_{t,h}, \; \mathcal{Z}_{t,h} \leftarrow \text{Update}(\mathbf{X}_{t,h}, \mathbf{Y}_{t,h})$     // *update model and maximizer candidate set*
    $\hat{d}_{t,h+1}(x) \leftarrow \hat{d}_{t,h} \oplus \delta(x_{t,h+1}, x)$    // *update empirical state-action distribution, see eq.* (11)
    **if** feedback is episodic **then**
      $\mathbf{Y}_{t,H} = f(\mathbf{X}_{t,H}) + \vec{\epsilon}_{t,:}$         // *obtain observations*
      $\mathcal{GP}_{t+1,:}, \; \mathcal{Z}_{t+1,:} \leftarrow \text{Update}(\mathbf{X}_{t,H}, \mathbf{Y}_{t,H})$    // *update model and maximizer candidate set*
  **Return:** Estimate of the maximum using the GP posterior's mean $\hat{x}_* = \arg\max_{x \in \mathcal{X}} \mu_T(x)$

---

**Set of potential maximizers $\mathcal{Z}$.** The definition of the objective requires the use of a set of maximizers. In the ideal case, we can say a particular input $x$, is not the optimum if there exists $x'$ such that $f(x') > f(x)$ with high confidence. We formalize this using the GP credible sets (Bayesian confidence sets) and define:

$$\mathcal{Z}_t = \{x \in \mathcal{X} : \text{UCB}(f(x)|\mathbf{X}_t) \geq \sup_{x' \in \mathcal{X}} \text{LCB}(f(x')|\mathbf{X}_t)\} \tag{9}$$

where UCB and LCB correspond to the upper and lower confidence bounds of the GP surrogate with a user specified confidence level defined via the posterior GP with data up to $\mathbf{X}_t$.

### 3.2 Discrete vs Continuous MDPs.

Until this point, our formulation focused on discrete $\mathcal{S}$ and $\mathcal{A}$ for ease of exposition. However, the framework is compatible with continuous state-action spaces. The probabilistic reformulation of the objective in Eq. (7) is possible irrespective of whether $\mathcal{X}$ (or $\mathcal{A}$) is a discrete or continuous subset of $\mathbb{R}^d$. In fact, the convexity of the objective in the space of distributions is still maintained. The difference is that the visitations $d$ are no longer probability mass functions but have to be expressed as probability density functions $d_c(x, a)$. To recover probabilities in the definition of $\mathbf{V}$, we need to replace sums with integrals i.e. $\sum_{x \in \mathcal{X}, a \in \mathcal{A}} d(x) \frac{\Phi(x,a)\Phi(x,a)^\top}{\sigma(x,a)^2} \rightarrow \int_{x \in \mathcal{X}, a \in \mathcal{A}} d_c(x, a) \frac{\Phi(x,a)\Phi(x,a)^\top}{\sigma(x,a)^2}$.

In the Eq. (8) we need to approximate a maximum over all input pairs in $\mathcal{Z}$. While this can be enumerated in the discrete case without issues, it poses a non-trivial constrained optimization problem when $\mathcal{X}$ is continuous. As an alternative, we propose approximating the set $\mathcal{Z}$ using a finite approximation of size $K$ which can be built using Thompson Sampling [57, 58] or through maximization of different UCBs for higher exploitation (see Appendix E.1). In Appendix E.5, we numerically benchmark reasonable choices of $K$, and show that the performance is not significantly affected by them.

### 3.3 General algorithm and Theory

The general algorithm combines the ideas introduced so far. We present it in Algorithm 1. Notice that apart from constructing the current utility, keeping track of the visited states and updating our GP model, an essential step is *planning*, where we need to find a policy that maximizes the utility. As this forms the core challenge of the algorithm, we devote Sec. 4 to it. In short, it solves a sequence of dynamic programming problems defined by the steps of the Frank-Wolfe algorithm. From a theoretical point of view, under the assumption of episodic feedback, the algorithm provably minimizes the utility as we show in Proposition C.1 in Appendix C.4.

## 4 Solving the planning problem

The planning problem, defined as $\min_{d_\pi \in \mathcal{D}} \mathcal{U}(d_\pi)$, can be thought of as analogous to optimizing an acquisition function in traditional BayesOpt, with the added difficulty of doing it in the space of

policies. See the bottom half of Figure 5 in Appendix A for a breakdown of the different components of our solution. Following developments in Hazan et al. [34] and Mutný et al. [17], we use the classical Frank-Wolfe algorithm [59]. It proceeds by decomposing the problem into a series of linear optimization sub-problems. Each linearization results in a policy, and we build a mixture policy consisting of optimal policies for each linearization $\pi_{\mathrm{mix},n} = \{(\alpha_i, \pi_i)\}_{i=1}^n$, and $\alpha_i$ step-sizes of Frank-Wolfe. Conveniently, after the linearization of $\mathcal{U}$ the subproblem on the polytope $\mathcal{D}$ corresponds to an RL problem with reward $\nabla \mathcal{U}$ for which many efficient solvers exist. Namely, for a single mixture component we have,

$$d_{\pi_{n+1}} = \arg\min_{d \in \mathcal{D}} \sum_{x,a,h} \nabla \mathcal{U}(d_{\pi_{\mathrm{mix},n}})(x,a) d_h(x,a). \tag{10}$$

Due to convexity, the state-action distribution follows the convex combination, $d_{\pi_{\mathrm{mix},n}} = \sum_{i=1}^n \alpha_i d_{\pi_i}$. The optimization produces a Markovian policy due to the subproblem in Eq. (10) being optimized by one. We now detail how to construct a non-Markovian policies by adaptive resampling.

### 4.1 Adaptive Resampling: Non-Markovian policies.

A core contribution of our paper is receding horizon re-planning. This means that we keep track of the past states visited in the *current* and *past* trajectories and adjust the policy at every step $h$ of the horizon $H$ in each trajectory indexed by $t$. At $h$, we construct a Markov policy for a reward that depends on all past visited states. This makes the resulting policy history dependent. While in episode $t$ and time-point $h$ we follow a Markov policy for a single step, the overall policy is a history-dependent non-Markov policy.

We define the empirical state-action visitation distribution,

$$\hat{d}_{t,h} = \frac{1}{tH + h} \Big( \underbrace{\sum_{j=1}^t \sum_{x,a \in \tau_j} \delta_{x,a}}_{\text{visited states in past trajectories}} + \underbrace{\sum_{x,a \in \tau_t|_h} \delta_{x,a}}_{\text{states at ep. } t \text{ up to } h} \Big) \tag{11}$$

where $\delta_{x,a}$ denotes a delta mass at state-action $(x,a)$. Instead of solving the objective $\mathcal{U}(d)$ as in Eq. (10), we seek to find a correction to the empirical distribution by minimizing,

$$\mathcal{U}_{t,h}(d) = \mathcal{U}\left(\frac{1}{H}\left(\frac{H-h}{1+t}d + \frac{tH+h}{1+t}\hat{d}_{t,h}\right)\right). \tag{12}$$

We use the same Frank-Wolfe machinery to optimize this objective: $d_{\pi_{t,h}} = \arg\min_{d_\pi \in \tilde{\mathcal{D}}} \mathcal{U}_{t,h}(d_\pi)$. The distribution $d_{\pi_{t,h}}$ represents the density of the policy to be deployed at trajectory $t$ and horizon counter $h$. We now need to solve multiple ($n$ due to FW) RL problems at each horizon counter $h$. Despite this, for discrete MDPs, the sub-problem can be solved extremely efficiently to exactness using dynamic programming. As can be seen in Appendix B.4, our solving times are just a few seconds, even if planning for very long horizons. The resulting policy $\pi$ can be found by marginalization $\pi_h(a|x) = d_{\pi,h}(x,a)/\sum_a d_{\pi,h}(x,a)$, a basic property of MDPs [60].

### 4.2 Continuous MDPs: Model Predictive Control

With continuous search space, the sub-problem can be solved using continuous RL solvers. However, this can be difficult. The intractable part of the problem is that the distribution $d_\pi$ needs to be represented in a computable fashion. We represent the distribution by the sequence of actions taken $\{a_h\}_{h=1}^T$ with the linear state-space model, $x_{h+1} = Ax_h + Ba_h$. While this formalism is not as general as it could be, it gives us a tractable sub-problem formulation common to control science scenario [61] that is practical for our experiments and captures a vast array of problems. The optimal set of actions is solved with the following problem, where we state it for the full horizon $H$:

$$\arg\min_{a_0,\ldots,a_H} \sum_{h=0}^H \nabla \mathcal{U}_{t,0}(d_{\pi_{\mathrm{mix},t}})(x_h, a_h), \tag{13}$$

such that $\|a_h\| \leq a_{\max}, x_h \in \mathcal{X}$, and $x_{h+1} = Ax_h + Ba_h$, where the *known* dynamics serves as constraints. Notice that instead of optimizing over the policy $d_\pi$, we directly optimize over the parameterizations of the policy $\{a_h\}_{h=1}^H$. In fact, this formulation is reminiscent of the model predictive control (MPC) optimization problem. Conceptually, these are the same. The only caveat in our case is that unlike in MPC [62], our objective is non-convex and tends to focus on gathering information rather than stability. Due to the non-convexity in this parameterization, we need to solve it heuristically. We identify a number of useful heuristics to solve this problem in Appendix G.

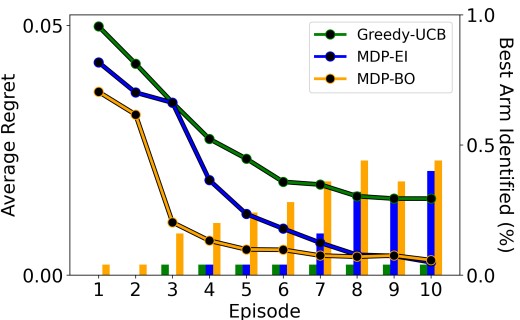 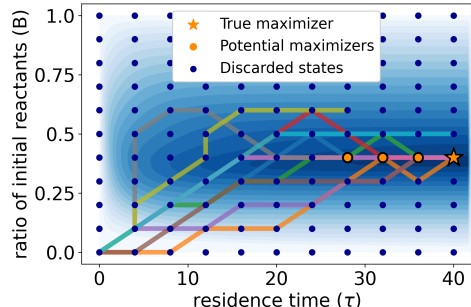

Figure 2: The Knorr pyrazole synthesis experiment. On the left, we show the quantitative results. The line plots denote the best prediction regret, while the bar charts denote the percentage of runs that correctly identify the best arm at the end of each episode. On the right, we show ten paths in different colours chosen by the algorithm. The underlying black-box function is shown as the contours, and we can see the discretization as dots. We can see four remaining potential maximizers (in orange), which includes the true one (star). *Notice all paths are non-decreasing in residence time, following the transition constraints.*

## 5 Experiments

Sections 5.1 – 5.3 showcase real-world applications under physical transitions constraints, using the discrete version of the algorithm. Section 5.4 benchmarks against other algorithms in the continuous setting, where we consider the additive transition model of Section 4.2 with $A = B = \mathbf{I}$. We include additional results in Appendix B. For each benchmark, we selected reasonable GP hyper-parameters and fixed them during the optimization. These are summarized in Appendix E.2. As we are interested maximizer identification, in discrete problems, we report the proportion of reruns that succeed at identifying the true maximum. For continuous benchmarks, we report inference regret at each iteration: $\text{Regret}_t = f(x_*) - f(x_{\mu,t})$, where $x_{\mu,t} = \arg\max_{x \in \mathcal{X}} \mu_t(x)$. All statistics reported are over 25 different runs.

**Baselines.** We include a naive baseline that greedily optimizes the immediate reward to showcase a method with no planning (Greedy-UCB). Likewise, we create a baseline that replaces the gradient in Eq. (10) with Expected Improvement [63] (MDP-EI), a weak version of planning. In the continuous settings, we compare against truncated SnAKe (TrSnaKe) [42], which minimizes movement distance, and against local search region-constrained BayesOpt or LSR [22] for the same task. We compare two variants for approximating the set of maximizers, one using Thompson Sampling (MDP-BO-TS) and one using Upper Confidence Bound (MDP-BO-UCB).

### 5.1 Knorr pyrazole synthesis

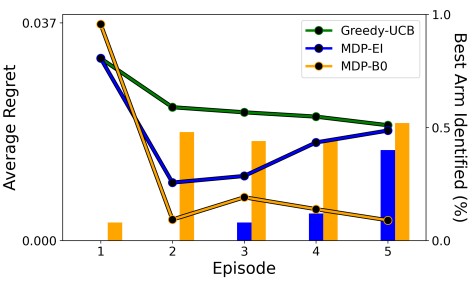 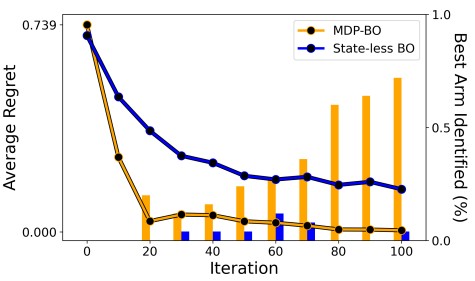

(a) Monitoring Lake Ypacarai.        (b) Free-electron laser tuning.

Figure 3: Results for Ypacarai and free electron-laser tuning experiments. On the left, the line plots denote the best prediction regret, while the bar charts denote the percentage of runs that correctly identify the best arm at the end of each episode. On the right, We plot the regret and compare against standard BO without accounting for movement-dependent noise.

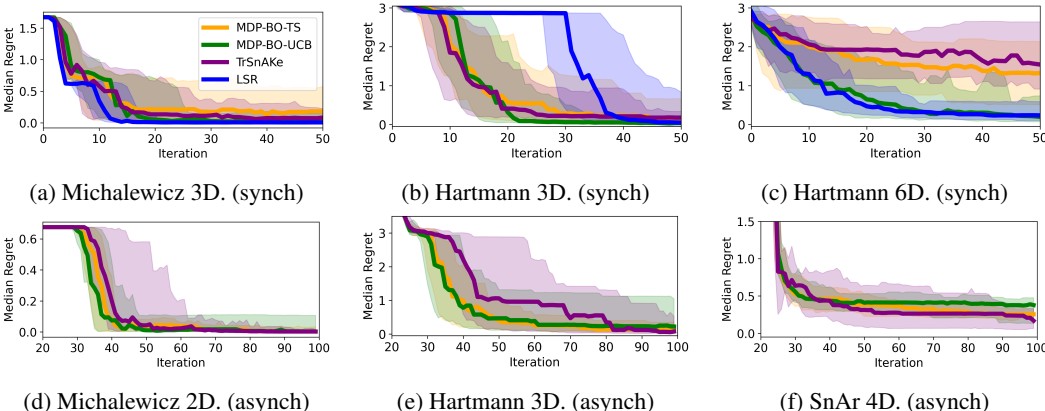

| | | |
|---|---|---|
| (a) Michalewicz 3D. (synch) | (b) Hartmann 3D. (synch) | (c) Hartmann 6D. (synch) |
| (d) Michalewicz 2D. (asynch) | (e) Hartmann 3D. (asynch) | (f) SnAr 4D. (asynch) |

Figure 4: Results of experiments on the asynchronous and synchronous benchmarks. We plot the median predictive regret and the 10% and 90% quantiles. For the asynchronous experiments, we can see that the paths taken by MDP-BO-TS are more consistent, and the final performance is comparable to TrSnAKe. While in the asynchronous setting, we found creating the maximization set using Thompson Sampling gave a stronger performance, in the synchronous setting, UCB is preferred. LSR gives a very strong performance, comparable to MDP-BO-UCB in almost all benchmarks.

Our chemical reactor benchmark synthetizes Knorr pyrzole in a transient flow reactor. In this experiment, we can control the flow-rate (residence time) $\tau$ and ratio of reactants $B$ in the reactor. We observe product concentration at discrete time intervals and we can also change inputs at these intervals. Our goal is to find the best parameters of the reaction subject to natural movement constraints on $B$, and $\tau$. In addition, we assume *decreasing* the flow rate of a reactor can be easily achieved. However, *increasing* the flow rate can lead to inaccurate readings [64]. A lower flow rate leads to higher residence time, so we impose that $\tau$ must be non-decreasing.

**The kernel.** Schrecker et al. [27] indicate the reaction can be approximately represented by simple kinetics via a differential equation model. We use this information along with techniques for representing linear ODE as constraints in GP fitting [65, 66] to create an approximate ODE kernel $k_{ode}$ through the featurization:

$$\Phi_{ode}(\tau, B) = (1 - \mathcal{S}(B))y^{(1)}(\tau, B) + \mathcal{S}(B)y^{(2)}(\tau, B)$$

where $y^{(i)}(\tau, B)$ are equal to:

$$\gamma_i(B)\left(\frac{\lambda_2^{(i)}}{\lambda_1^{(i)} - \lambda_2^{(i)}}e^{\lambda_1^{(i)}\tau} - \frac{\lambda_1^{(i)}}{\lambda_1^{(i)} - \lambda_2^{(i)}}e^{\lambda_2^{(i)}\tau} + 1\right)$$

for $i = 1, 2$, where $\lambda_1^{(i)}$ and $\lambda_2^{(i)}$ are eigenvalues of the linearized ODE at different stationary points, $\gamma_1(B) = B$, $\gamma_2(B) = 1 - B$, and $\mathcal{S}(x) := (1 + e^{-\alpha_{sig}(x-0.5)})^{-1}$ is a sigmoid function. Appendix H holds the details and derivations which may be of independent interest. As the above kernel is only an approximation of the true ODE kernel, which itself is imperfect, we must account for the model mismatch. Therefore, we add a squared exponential term to the kernel to ensure a non-parametric correction, i.e.: $k(\tau, B) = \alpha_{ode}k_{ode}(\tau, B) + \alpha_{rbf}(\tau, B)$.

We report the examples of the trajectories in the search space in Figure 2. Notice that all satisfy the transition constraints. The paths are not space-filling and avoid sub-optimal areas because of our choice of non-isotropic kernel based on the ODE considerations. We run the experiment with episodic feedback, for 10 episodes of length 10 each, starting each episode with $(\tau_R, B) = (0, 0)$. Figure 2 reports quantitative results and shows that the best-performing algorithm is MDP-BO.

## 5.2 Monitoring Lake Ypacarai

Samaniego et al. [20] investigated automatic monitoring of Lake Ypacarai, and Folch et al. [11] and Yang et al. [40] benchmarked different BayesOpt algorithms for the task of finding the largest contamination source in the lake. We introduce local transition constraints to this benchmark by creating the lake containing obstacles that limit movement (see Figure 12 in the Appendix). Such obstacles

in environmental monitoring may include islands or protected areas for animals. We add an initial and final state constraint with the goal of modeling that the boat has to finish at a maintenance port.

We focus on *episodic* feedback, where each episode consists of 50 iterations. Results can be seen in Figure 3a. MDP-EI struggles to identify the maximum contamination for the first few episodes. On the other hand, our method correctly identifies the maximum in approximately 50% of the runs by episode two and achieves better regret.

### 5.3 Free-electron laser: Transition-driven corruption

Apart from hard constraints, we can apply our framework to state-dependent BayesOpt problems involving transitions. For example, the magnitude of noise $\epsilon$ may depend on the transition. This occurs in systems observing equilibration constraints such as a free-electron laser [23]. Using the simplified simulator of this laser [67], we use our framework to model heteroscedastic noise depending on the difference between the current and next state, $\sigma^2(x, x') = s(1 + w||x - x'||_2)$. By choosing $\mathcal{A} = \mathcal{X}$, we rewrite the problem as $\sigma(s, a) = s(1 + w||x - a||_2)$. The larger the move, the more noisy the observation. This creates a problem, where the BayesOpt needs to balance between informative actions and movement, which can be directly implemented in the objective (8) via the matrix $\mathbf{V}(d_\pi) = \sum_{x,a \in \mathcal{X}} d_\pi(x, a) \frac{1}{\sigma^2(x,a)} \Phi(x)\Phi(x)^\top + \frac{1}{TH}\mathbf{I}$. Figure 3b reports the comparison between worst-case stateless BO and our algorithm. Our approach substantially improves performance.

### 5.4 Synthetic Benchmarks

We benchmark on a variety of classical BayesOpt problems while imposing local movement constraints and considering both immediate and asynchronous feedback (by introducing an observation delay of 25 iterations). We also include the chemistry SnAr benchmark, from Summit [68], which we treat as asynchronous as per Folch et al. [11]. Results are in Figure 4. In the synchronous setting, we found using the UCB maximizer criteria for MDP-BO yields the best results (c.f. Appendix for details of this variant). We also found that LSR performs very competitively on many benchmarks, frequently matching the performance of MDP-BO. In the asynchronous settings we achieved better results using MDP-BO with Thompson sampling. TrSnAKe baseline appears to be competitive in all synthetic benchmarks as well. However, MDP-BO is more robust having less variance in the chosen paths as seen in the quantiles. It is important to highlight that SnAKe and LSR are specialist heuristic algorithms for local box-constraints, and therefore it is not surprising they perform strongly. Our method can be applied to more general settings and therefore it is very encouraging that MDP-BO is able to match these SOTA algorithms in their specialist domain.

## 6 Conclusion

We considered transition-constrained BayesOpt problems arising in physical sciences, such as chemical reactor optimization, that require careful planning to reach any system configuration. Focusing on maximizer identification, we formulated the problem with transition constraints using the framework of Markov decision processes and constructed a tractable algorithm for provably and efficiently solving these problems using dynamic programming or model predictive control sub-routines. We showcased strong empirical performance in a large variety of problems with physical transitions, and achieve state-of-the-art results in classical BayesOpt benchmarks under local movement constraints. This work takes an important step towards the larger application of Bayesian Optimization to real-world problems. Further work could address the continuous variant of the framework to deal with more general transition dynamics, or explore the performance of new objective functions.

## Acknowledgments

JPF is funded by EPSRC through the Modern Statistics and Statistical Machine Learning (StatML) CDT (grant no. EP/S023151/1) and by BASF SE, Ludwigshafen am Rhein. RM acknowledges support from the BASF / Royal Academy of Engineering Research Chair in Data-Driven Optimisation. This publication was created as part of NCCR Catalysis (grant number 180544), a National Centre of Competence in Research funded by the Swiss National Science Foundation, and was partially supported by the European Research Council (ERC) under the European Union's Horizon 2020 research and Innovation Program Grant agreement no. 815943. We would also like to thank Linden Schrecker, Ruby Sedgwick, and Daniel Lengyel for providing valuable feedback on the project.

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

# A  Visual abstract of the algorithm

In Figure 5 we summarize how our algorithm creates non-Markovian policies for maximizer identification and the corresponding connections to other works in the literature.

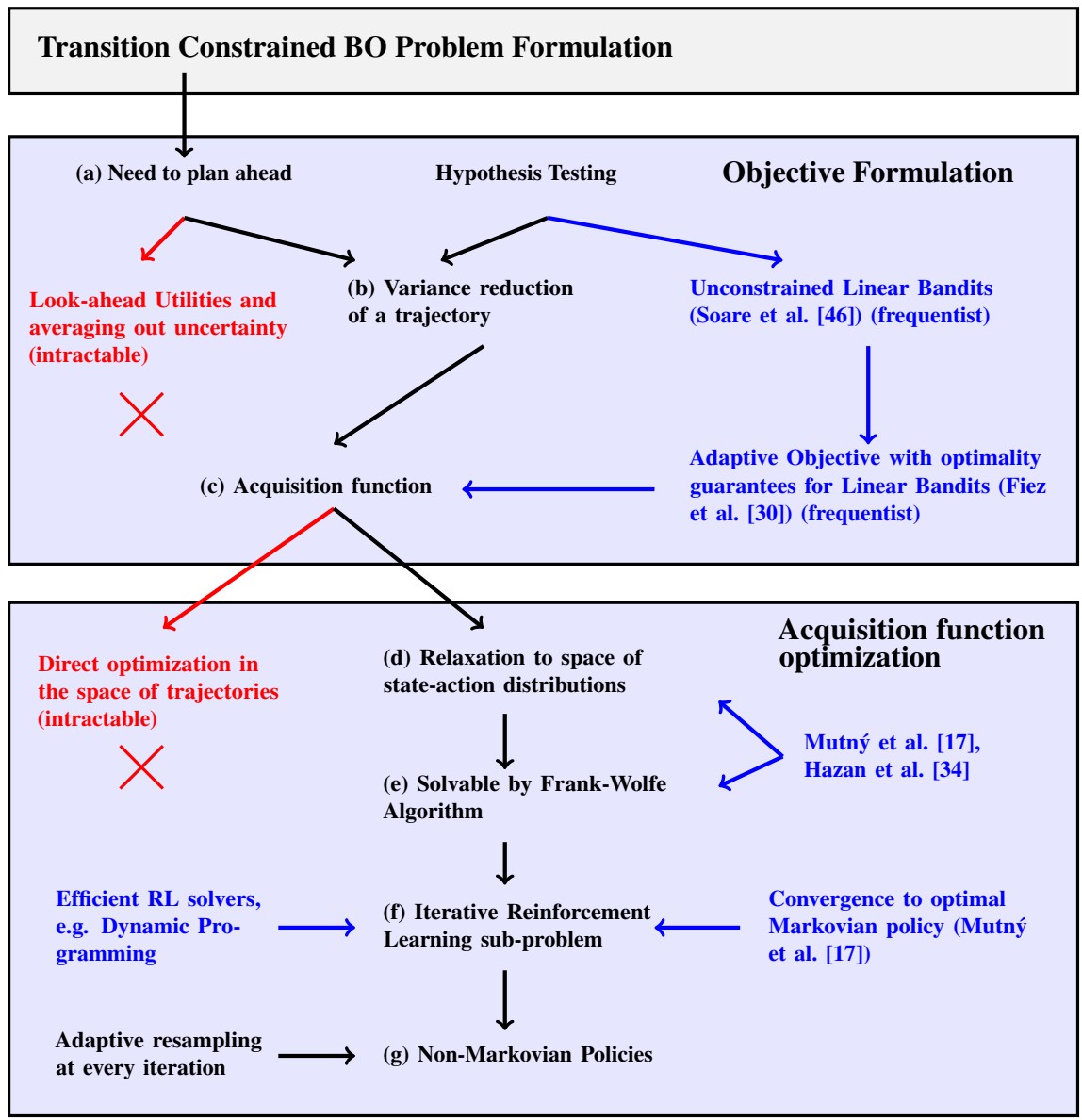

Figure 5: Visual abstract of the work. In black we show the method presented in this paper, with literature connections shown in blue. In red we show solutions which we did not pursue due to intractability. The problem creates the **(a) need to plan ahead**. To do this, we take inspiration from hypothesis testing and focus on **(b) the variance reduction in a set of maximizers**, which leads to our **(c) acquisition function**. The objective is the same as Fiez et al. [30] introduced in the linear bandits literature from a frequentist perspective. To optimize it, we follow developments in Mutný et al. [17], Hazan et al. [34] by **(d) relaxing the acquisition function to the space of state-action distributions** and **(e) solving the planning problem using the Frank-Wolfe algorithm**. This consists of iteratively solving tractable **(f) reinforcement learning sub-problems** which give us optimal Markov policies. We then apply adaptive resampling to obtain **(g) non-Markovian policies**.

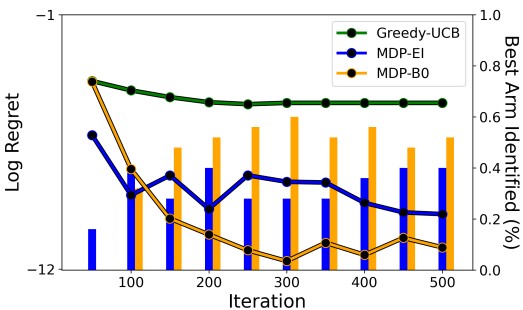

Figure 6: High noise constrained Ypacari experiment with immediate feedback.

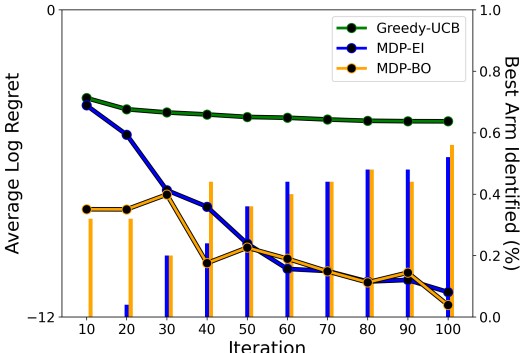

Figure 7: Knorr pyrazole synthesis with immediate feedback

# B  Additional Empirical Results

### B.1  Constrained Ypacarai

We also run the Ypacarai experiment with immediate feedback. To increase the difficulty, we used large observation noise, $\sigma^2 = 0.01$. The results can be seen in Figure 6. The early performance of MDP-EI is much stronger, however, it gets overtaken by our algorithm from episode three onwards, and gives the worst result at the end, as it struggles to identify which of the two optima is the global one.

### B.2  Knorr pyrazole synthesis

We also include results for the Knorr pyrazole synthesis with immediate feedback. In this case we observe very strong early performance from MDP-BO, but by the end MDP-EI is comparable. The greedy method performs very poorly.

### B.3  Additional synthetic benchmarks

Finally, we also include additional results on more synthetic benchmarks for both synchronous and asynchronous feedback. The results are shown in Figures 8 and 9. The results back the conclusions in the main body. All benchmarks do well in 2-dimensions while highlighting further that MDP-BO-UCB and LSR can be much stronger in the synchronous setting than Thompson Sampling planning-based approaches (with the one exception of the Levy function).

Table 1: Average acquisition function solving times for each practical benchmark. We give the solving times to the nearest second, and provide the size of the state-space, $|\mathcal{S}|$, the maximum number of actions one can take from a specific state, $|\mathcal{A}(S)|$, and the planning horizon. In all benchmarks we are able to solve the problem in a few seconds.

| Benchmark | Solve Time | $|\mathcal{S}|$ | Maximum $|\mathcal{A}(S)|$ | Planning horizon |
|---|---|---|---|---|
| Knorr pyrazole | 1s | 100 | 6 | 10 |
| Ypacarai | 3s | 100 | 8 | 50 |
| Electron laser | 15s | 100 | 100 | 100 |

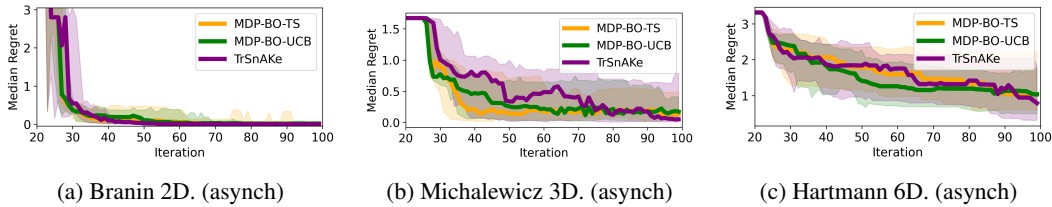

(a) Branin 2D. (asynch)  (b) Michalewicz 3D. (asynch)  (c) Hartmann 6D. (asynch)

Figure 8: Additional asynchronous results.

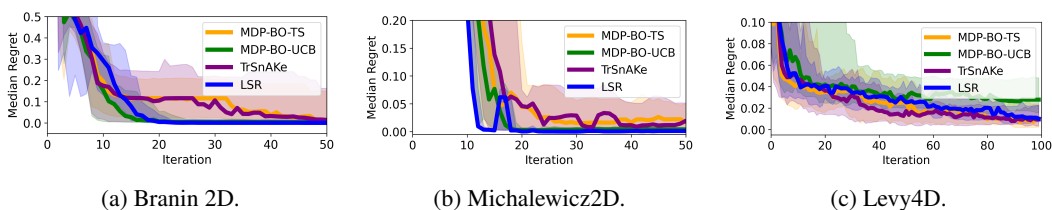

(a) Branin 2D.  (b) Michalewicz2D.  (c) Levy4D.

Figure 9: Additional synchronous results.

## B.4 Computational study

We include the average acquisition function solving time for each of the discrete problems. For the continuous case the running time was comparable to Truncated SnAKe [42] since most of the computational load was to create the set of maximizers using Thompson Sampling. The times were obtained in a simple 2015 MacBook Pro 2.5 GHz Quad-Core Intel Core i7. The bulk of the experiments was ran in parallel on a High Performance Computing cluser, equipped with AMD EPYC 7742 processors and 16GB of RAM.

## B.5 Median plots for Ypacarai and reactor experiments

In Figures 10 and 11 we give the median and quantile plots for the Knorr pyrazole synthesis and the Ypacarai experiment, which were not included in the main paper to avoid cluttering the graphics.

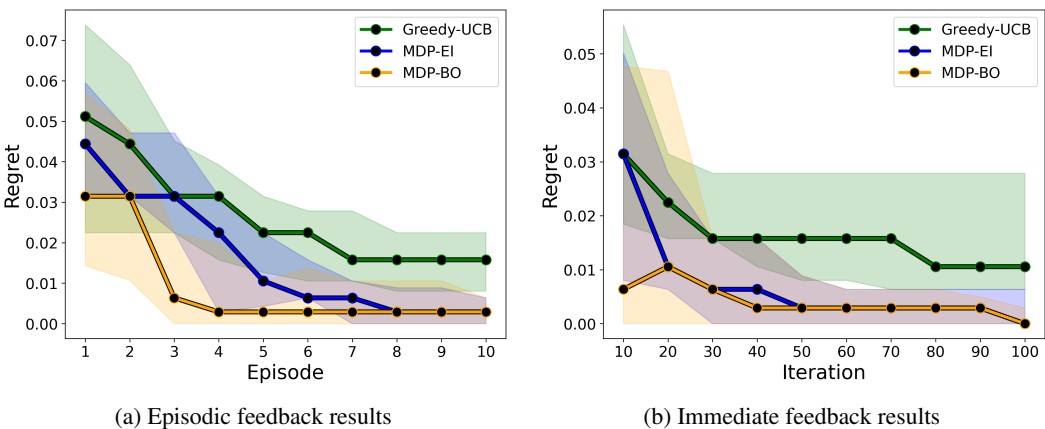

(a) Episodic feedback results  (b) Immediate feedback results

Figure 10: Median and 10th/90th quantile plots for Knorr pyrazole synthesis experiment.

## C  Utility function: Additional Info

We describe the utility function in complete detail using the kernelized variant that allows to extend the utility beyond the low-rank assumption in the main text.

### C.1  Derivation of the Bayesian utility

Suppose that our decision rule is to report the best guess of the maximizer after the $T$ steps as,

$$x_T = \arg\max_{x \in \mathcal{Z}} \mu_T(x).$$

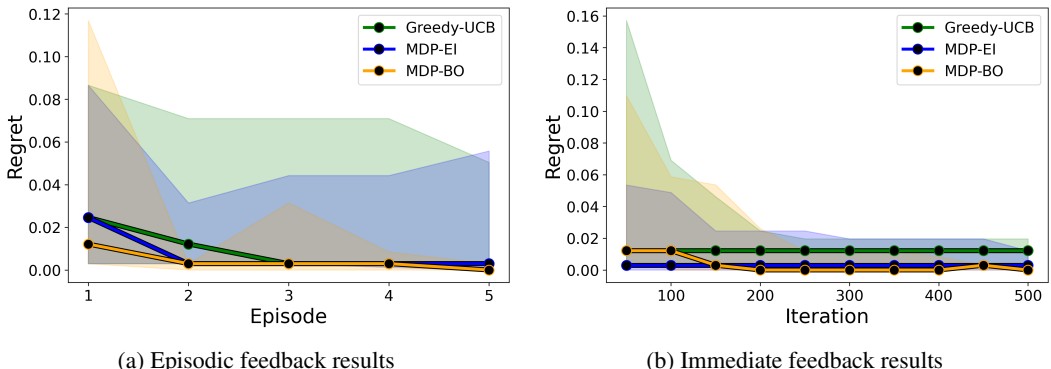

(a) Episodic feedback results      (b) Immediate feedback results

Figure 11: Median and 10th/90th quantile plots for Ypacarai experiment.

We call this the selection the *recommendation rule*. We focus on this recommendation rule as this rule is interpretable to the facilitator of the analysis and experimenters. In this derivation we use that $f = \theta^\top \Phi(x)$. More commonly, the notation $\langle \theta, \Phi(x) \rangle$ is used, where the inner product is potentially infinite dimensional. We use use $\top$ notation for simplicity for both cases. Same is true for any other functional estimates, e.g., for the posterior mean estimate, we use $\mu_t(x) = \Phi(x)^\top \mu_t$. The inner product is in the reproducing kernel Hilbert space associated with the kernel $k$.

Now, suppose there is a given $f$ (we will take expectation over it later), then there is an $x \in \mathcal{X}$ achieving optimum value, denoted $x^\star_f$ (suppose unique for this development here). Hence, we would like to model the risk associated with predicting a fixed $z \neq x^\star_f$, which is still in $\mathcal{Z}$ at time $T$. Suppose we are at time $t$, we develop the utility to gather additional data $\mathbf{X}_{\text{new}}$ on top of the already acquired data $\mathbf{X}_t$. These should improve the discrepancy of the true answer, and the reported value the most.

Suppose there are two elements in $\mathcal{Z}_{\text{simple}} = \{z, x^\star_f\}$. We will generalize to a composite hypothesis later. In two-element case, the probability of the error in incurred due to selecting $z$ is:

$$P(\mu_T(z) - \mu_T(x^\star_f) \geq 0 | f)$$

. The randomness here is due to the observations $y = f(X_{\text{new}}) + \epsilon$ that are used to fit the estimator $\mu_T(x)$. Namely due to $\epsilon \sim \mathcal{N}(0, \sigma^2)$. Given $f$ (equivalently $\theta$), the distribution of our estimator (namely the posterior mean) is Gaussian. Hence, given $f$:

$$\mu_T \sim \mathcal{N}((\mathbf{V}_T + \mathbf{I}_\mathcal{H})^{-1}\mathbf{V}_T\theta, \sigma^2(\mathbf{V}_T + \mathbf{I}_\mathcal{H})^{-1}V_T(\mathbf{V}_T + \mathbf{I}_\mathcal{H})^{-1}),$$

where $\mathbf{V} = \sum_{i=1}^T \frac{1}{\sigma^2}\Phi(x_i)\Phi(x_i)^\top$ is an operator on the reproducing kernel Hilbert space due to $k$ as $\mathcal{H} \to \mathcal{H}$, and $\mathbf{I}_\mathcal{H}$ the identity operator on the same space.

This is the posterior over the posterior mean as a function. A posterior over the specific evaluation is $\mu_T(z) - \mu_T(x^\star_f) \sim$ $\mathcal{N}(\underbrace{\theta^\top(\mathbf{V}_T + \mathbf{I}_\mathcal{H})^{-1}\mathbf{V}_T(\Phi(z) - \Phi(x^\star_f))}_{a}, \underbrace{\sigma^2(\Phi(z) - \Phi(x^\star_f))^\top(\mathbf{V}_T + \mathbf{I}_\mathcal{H})^{-1}\mathbf{V}_T(\mathbf{V}_T + \mathbf{I}_\mathcal{H})^{-1}(\Phi(z) - \Phi(x^\star_f))}_{b^2}).$

We can now bound the probability of making an error using a Gaussian tail bound inequality:

$$\mathbb{P}(\mu_T(z) - \mu_T(x^*) \geq 0) = \mathbb{P}(\mu_T(z) - \mu_T(x^*) \geq a_z + (-a_z)) \leq e^{-\frac{a^2}{2b^2}}$$

with the caveat that the inequality only holds when the $a_z$ is negative. However note that $a_z \to f(z) - f(x^*) < 0$ as $T \to \infty$ therefore it will hold once $T$ is large enough. From this we can take logarithms and then the expectation across the randomness in the GP:

$$\mathbb{E}_{f \sim GP}[\log \mathbb{P}(\mu_T(z) - \mu_T(x^*)|f)] \leq -\frac{1}{2}\mathbb{E}_{f \sim GP}\left[\frac{a_z^2}{b_z^2}\right]$$

which is called the log *Bayes' factor* and is expected log failure rate for the set of potential maximizers $\mathcal{Z}_{\text{simple}}$. The expectation is over the posterior including the evaluations $\mathbf{X}_t$ (or prior at the very beginning of the procedure). In fact, we can think of the posterior as being the new prior for the future

at any time point. Now assuming that $\mathcal{Z}$ has more than one additional element, we want to ensure the failure rate is small for all other failure modes, all other hypothesis. Technically this means, we have an alternate hypothesis, which is *composite* – composed of multiple point hypotheses. We take the worst-case perspective as its common with composite hypotheses. In expectation over the prior, we want to minimize:

$$\min_{\mathbf{X}_{\text{next}}} \mathbb{E}_f \left[ \sup_{z \in \mathcal{Z} \setminus \{x_f^\star\}} \log P(\mu_T(z) - \mu_T(x_f^\star) \geq 0|f) \right]. \tag{14}$$

For moderate to large $T \gg 0$, we can upper bound this objective via elegant argument to yield a very transparent objective:

$$\min_{\mathbf{X}_{\text{next}}} \mathbb{E}_f \left[ \sup_{z \in \mathcal{Z} \setminus \{x_f^\star\}} \log P(\mu_T(z) - \mu_T(x_f^\star) \geq 0|f) \right] \dot{\leq} -\frac{1}{2} \min_{\mathbf{X}_{\text{next}}} \mathbb{E}_f \left[ \sup_{z \in \mathcal{Z} \setminus \{x_f^\star\}} \frac{(f(z) - f(x_f^\star))^2}{k_{\mathbf{X}_t \cup \mathbf{X}_{\text{new}}}(z, x_f^\star)} \right] \tag{15}$$

where we have used an lower and upper bound on the $a_z$ and $b_z$, respectively as follows:

$$
\begin{aligned}
a_z^2 &= (\theta^\top (\mathbf{V}_T + \mathbf{I}_\mathcal{H})^{-1} \mathbf{V}_T (\Phi(z) - \Phi(x_f^\star)))^2 \\
&= (\theta^\top (\mathbf{V}_T + \mathbf{I}_\mathcal{H})^{-1} (\mathbf{V}_T + \mathbf{I}_\mathcal{H} - \mathbf{I}_\mathcal{H})(\Phi(z) - \Phi(x_f^\star)))^2 \\
&= (\theta^\top (\Phi(z) - \Phi(x_f^\star)) - \theta^\top (\mathbf{V}_T + \mathbf{I}_\mathcal{H})^{-1} (\Phi(z) - \Phi(x_f^\star)))^2 \\
&\stackrel{T \gg 0}{\approx} (\theta^\top (\Phi(z) - \Phi(x_f^\star)))^2 = (f(z) - f(x_f^\star))^2 \\
b_z^2 &= \sigma^2 (\Phi(z) - \Phi(x_f^\star))^\top (\mathbf{V}_T + \mathbf{I}_\mathcal{H})^{-1} \mathbf{V}_T (\mathbf{V}_T + \mathbf{I}_\mathcal{H})^{-1} (\Phi(z) - \Phi(x_f^\star)) \\
&\leq \sigma^2 (\Phi(z) - \Phi(x_f^\star))^\top (\mathbf{V}_T + \mathbf{I}_\mathcal{H})^{-1} (\Phi(z) - \Phi(x_f^\star)) = k_{\mathbf{X}}(z, x_f^\star).
\end{aligned}
$$

In the last line we have used the same identity as in Eq. (25). We will explain how to eliminate the expectation in Section C.2

### C.2   Upper-bounding the objective: Eliminating $\mathbb{E}_f$ for large $T$.

The objective Eq. (15) is intractable due to the expectation of the prior and which involves expectation over the maximum $f(x_f^\star)$, which is known to be very difficult to estimate. Interestingly, the denominator is independent of $f$ if we adopt the worst-case perspective over the $x_f^\star$, and hence the only dependence is through the set $\mathcal{Z}$ as well as the denominator. Given all current prior information, we can determine $\mathcal{Z}$, and hence split the expectation. Let us now express

At any time point, we can upper-bound the denominator by the minimum as done by Fiez et al. [30]. Even if $\mathcal{Z}$ decreases, as we get more information, the worst-case bound is always proportional to the smallest gap $\text{gap}(f)$ between two arms in $\mathcal{X}$. Hence, we can upper bound the objective as:

$$
\begin{aligned}
-\mathbb{E}_f \left[ \sup_{z \in \mathcal{Z} \setminus \{x_f^\star\}} \frac{(f(z) - f(x_f^\star))^2}{k_{\mathbf{X} \cup \mathbf{X}_{\text{new}}}(z, x_f^\star)} \right] &\leq -\sup_{z \in \mathcal{Z} \setminus \{x_f^\star\}} \frac{\mathbb{E}_f [\text{gap}(f)]}{k_{\mathbf{X} \cup \mathbf{X}_{\text{new}}}(z, x_f^\star)} \\
&\leq -\mathbb{E}_f [\text{gap}(f)] \sup_{z \in \mathcal{Z} \setminus \{x_f^\star\}} \frac{1}{\text{Var} \left[ f(z) - f(x_f^\star) | \mathbf{X}_t \cup \mathbf{X}_{\text{new}} \right]}.
\end{aligned}
$$

As the constant in front of the objective does not influence the optimization problem, we do not need to consider it when defining the utility. Furthermore, in order to minimise the probability of an error we can just minimise the variance in the denominator instead (since $\arg \min_x -g(x)$ is equivalent to $\arg \min_x \frac{1}{g(x)}$ when $g(x) > 0$). However, the non-trivial distribution of $f(x^\star)$ [29] renders the utility intractable; therefore we employ a simple and tractable upper bound on the objective by minimising the uncertainty among *all* pairs in $\mathcal{Z}$:

$$U(\mathbf{X}_{\text{new}}) = \max_{z', z \in \mathcal{Z}, z \neq z'} \text{Var}[f(z) - f(z') | \mathbf{X}_t \cup \mathbf{X}_{\text{new}}]. \tag{16}$$

Surprisingly, this objective coincides with the objective from Fiez et al. [30] which has been derived as lower bound to the best-arm identification problem (maximum identification) with linear bandits. Their perspective is however slightly different as they try to minimize $T$ for a fixed $\delta$ failure rate. Perhaps it should not be surprising that the dual variant, consider here, for fixed $T$ and trying to minimize the failure rate leads to the same decision for large $T$ when $\log(b_z)$ can be neglected.

## C.3 Approximation of Gaussian Processes

Let us now briefly summarize the Nyström approximation [31, 69]. Given a kernel $k(\cdot, \cdot)$, and a data-set $X$, we can choose a sub-sample of the data $\hat{x}_1, ..., \hat{x}_m$. Using this sample, we can create a low $r$-rank approximation of the full kernel matrix

$$\hat{K}_r = K_b \hat{K}^\dagger K_b$$

where $K_b = [k(x_i, \hat{x}_j)]_{N \times m}$, $\hat{K} = [k(\hat{x}_i, \hat{x}_j)]_{m \times m}$ and $K^\dagger$ denotes the pseudo-inverse operation. We can then define the Nyström features as:

$$\phi_n(x) = \hat{D}_r^{-1/2} \hat{V}_r^T (k(x, x_1), ..., k(x, x_m))^T, \tag{17}$$

where $\hat{D}_r$ is the diagonal matrix of non-zero eigenvalues of $\hat{K}_r$ and $\hat{V}_r$ the corresponding matrix of eigenvectors. It follows that we obtain a finite-dimensional estimate of the GP:

$$f(x) \approx \Phi(x)^T \theta \tag{18}$$

where $\Phi(x) = (\phi_1(x), \dots \phi_m(x))^T$, and $\theta$ are weights with a Gaussian prior.

## C.4 Theory: convergence to the optimal policy

The fact that our objective is derived using Bayesian decision theory makes it well-rooted in theory. In addition to the derivation of Section C.1, we can prove that our scheme is able to converge in terms of the utility.

Notice that the set of potential maximizers is changing over time, and hence we add a time subscript to $\mathcal{Z}$ as $\mathcal{Z}_t$. Let us contemplate for a second what could the optimal policy. As the set of $\mathcal{Z}_t$ is changing, we follow the line of work of started by Russo [70] and introduce an optimal algorithm that knows the true $x_f^\star$ for each possible realization of the prior $f$. In other words, its an algorithm that any time $t$, would follow:

$$d_t^\star = \min_{d \in \mathcal{D}} \mathbb{E}_f \left[ \max_{z \in \mathcal{Z}_t \setminus \{x_f^\star\}} k_{\hat{d}_t \oplus d}(z, x_f^\star) \right],$$

where in the above $\hat{d}_t \oplus d$ represents the weighted sum as in the main Algorithm 1 that scales the distributions properly according to $t$ and $T$, so to make the sum of them a valid distribution. Notice that in contrast to our objective, it does not take the maximum over $z' \in \mathcal{Z}$, but fixes it to the value $x_f^\star$ that the hypothetical algorithm has privileged access to. To eliminate the cumbersome notation, we will refer to the objectives as $\mathcal{U}(d|\mathcal{Z}_t, \mathcal{Z}_t)$ as the objective used by our algorithm (real execution) and $\mathcal{U}(d|\mathcal{Z}_t, \{x_f^\star\})$, as the objective that the privileged algorithm is optimizing which serves as theoretical baseline.

The visitation of $d_t^\star$ represents the best possible investment of the resources (of the size $T - t$) to execute at time $t$ had we known the $x_f^\star$ instead of only $\mathcal{Z}_t$. This is interpreted as if the modeler knows $x_f^\star$, and sets up an optimal curriculum that is being shown to an observer in order to convince him/her of that $x_f^\star$ is the optimal value. He or she is using statistical testing to elucidate it from execution of the policy. Like the algorithm, the optimal policy changes along the optimization procedure due to changes in $\mathcal{Z}_t$. Hence, our goal is to show that we are closely tracking the performance of these optimal policies in time $t$, and eventually there is little difference between our sequence of executed policies (visitations) $\hat{d}_t$ and the algorithm optimal $d_t^\star$.

In order to prove the theorem formally, we need to assume that $\mathcal{Z}_t$ is decreasing. The rate at which this set is decreasing determines the performance of the algorithm to a large extent. Namely, we assume that given two points in time, having the same empirical information $\hat{d}_t$. Given, $f$, suppose

$$\sup_{d \in \mathcal{D}} |d^\top (\nabla \mathcal{U}(\hat{d}_t | \mathcal{Z}_t, \mathcal{Z}_t) - \nabla \mathcal{U}(\hat{d}_t | \mathcal{Z}_t, \{x_f^\top\}))| \leq C_t. \tag{$\star$}$$

As we gather information in our procedure the, $\{x_f^\star\} \subset \mathcal{Z}_t \subseteq \mathcal{Z}_{t-1}$, but the exact decrease depends on how $\mathcal{Z}_t$ is constructed. We leave the particular choice for $C_t$ to make the above hold for future work. We conjecture that this is decreasing as $C_t \approx \frac{\gamma_t}{\sqrt{t}}$, where $\gamma_t$ is the information gain due to Srinivas et al. [71]. We are now ready to state the formal theorem along with its assumptions.

**Proposition C.1.** *Assuming episodic feedback, and suppose that for any $\mathcal{Z}$,*

1. *$\mathcal{U}$ is convex on $\mathcal{D}$*

2. *B-locally Lipschitz continuous under $|| \cdot ||_\infty$ norm*

3. *locally smooth with constant L, i.e,*

$$\mathcal{U}(\eta + \alpha h) \leq \mathcal{U}(\eta) + \nabla\mathcal{U}(\eta)^\top h + \frac{L_{\eta,\alpha}}{2}\|h\|_2^2. \tag{19}$$

   *for $\alpha \in (0,1)$ and $\eta, h \in \Delta_p$, $L := \max_{\eta,\alpha} L_{\eta,\alpha}$*

4. *condition in $(\star)$ holds with Bayesian posterior inference,*

*we can show that the Algorithm 1 satisfied for the sequences of iterates $\{\hat{d}_t\}_{t=1}^T$:*

$$\frac{1}{T}\sum_{t=1}^{T-1}\mathcal{U}(d_t|\mathcal{Z}_t, \{x_f^\star\}) - \mathcal{U}(d_t^\star|\mathcal{Z}_t, \{x_f^\star\}) \leq \mathcal{O}\left(\frac{1}{T}\sum_{t=1}^{T-1}C_t + \frac{L\log T}{T} + \frac{B}{\sqrt{T}}\log\left(\frac{1}{\delta}\right)\right),$$

*with probability $1 - \delta$ on the sampling from the Markov chain. The randomness on the confidence set is captured by Assumption in Eq. $(\star)$.*

The previous proposition shows that as the budget of the experimental campaign $T$ is increasing, we are increasingly converging to the optimal allocation of the experimental resources on average also on the objective that is unknown to us. In other words, our algorithm is becoming approximately optimal also under the privileged information setting representing the best possible algorithm. Despite having a limited understanding of potential maximizers at the beginning by following our procedure, we show that we are competitive to the best possible allocation of the resources. Now, we prove the Proposition. The proof is an extension of the Theorem 3 in [17]. Whether the objective satisfied the above conditions depends on the set $\mathcal{X}$. Should the objective not satisfy smoothness, it can be easily extended by using the Nesterov smoothing technique as explained in the same priorly cited work.

*Proof of Proposition C.1.* The proof is based on the proof of Frank-Wolfe convergence that appears Appendix B.4 in Thm. 3. in Mutný et al. [17].

Let us start by notation. We will use the notation that $\mathcal{U}_t$ is the privileged objective $\mathcal{U}(d|\mathcal{Z}_t, \{x_t^f\})$, while the original objective will be specified as $\mathcal{U}(d|\mathcal{Z}_t, \mathcal{Z}_t)$.

First, what we follow in the algorithm:

$$q_t = \arg\min_{d\in\mathcal{D}} \nabla\mathcal{U}(\hat{d}_t|\mathcal{Z}_t, \mathcal{Z}_t)^\top d \tag{20}$$

The executed visitation is simply generated via sampling a trajectory from $q_t$. Let us denote the empirical visiation of the trajectory as $\delta_t$,

$$\delta_t \sim q_t. \tag{21}$$

For the analysis, we also need the best greedy step for the unknown (privileged) objective $\mathcal{U}$ as

$$z_t = \arg\min_{d\in\mathcal{D}} \nabla\mathcal{U}_t\left(\hat{d}_t\right)^\top d. \tag{22}$$

Let us start by considering the one step update:

$$\mathcal{U}_t(\hat{d}_{t+1}) \quad = \quad \mathcal{U}_t\left(\hat{d}_t + \frac{1}{t+1}(\delta_t - \hat{d}_t)\right)$$

$$\overset{L\text{-smooth}}{\leq} \quad \mathcal{U}_t(\hat{d}_t) + \frac{1}{t+1}\nabla\mathcal{U}_t(\hat{d}_t)^\top(\delta_t - \hat{d}_t) + \frac{L}{2(1+t)^2}\left\|\delta_t - \hat{d}_t\right\|^2$$

$$\mathcal{U}(\hat{d}_{t+1}) \quad \overset{\text{bounded}}{\leq} \quad \mathcal{U}_t(\hat{d}_t) + \frac{1}{t+1}\nabla\mathcal{U}_t(\hat{d}_t)^\top(\delta_t - \hat{d}_t) + \frac{L}{(1+t)^2}$$

$$= \quad \mathcal{U}_t(\hat{d}_t) + \frac{1}{t+1}\nabla\mathcal{U}_t(\hat{d}_t)^\top(q_t - \hat{d}_t) + \frac{1}{t+1}\underbrace{\nabla\mathcal{U}_t(\hat{d}_t)^\top(-q_t + \delta_t)}_{\epsilon_t} + \frac{L}{(1+t)^2}$$

We will now carefully insert and subtract two set of terms depending on the real objective so that we can bound them using $(\star)$:

$$
= \quad \mathcal{U}_t(\hat{d}_t) + \frac{1}{t+1}\nabla(\mathcal{U}_t(\hat{d}_t)^\top - \nabla\mathcal{U}_t(\hat{d}_t|\mathcal{Z}_t, \mathcal{Z}_t)^\top)(q_t - z_t) + \frac{1}{1+t}\mathcal{U}_t(\hat{d}_t)^\top(z_t - \hat{d}_t)
$$

$$
+ \frac{1}{1+t}\epsilon_t + \frac{L}{(1+t)^2}
$$

$$
\overset{\text{Using}\,\star}{\leq} \quad \mathcal{U}_t(\hat{d}_t) + 2\frac{1}{1+t}C_t + \frac{1}{t+1}\mathcal{U}_t(\hat{d}_t)^\top(z_t - \hat{d}_t) + \frac{1}{1+t}\epsilon_t + \frac{L}{(1+t)^2}
$$

Carrying on,

$$
\mathcal{U}_t(\hat{d}_{t+1}) \quad \overset{(22)}{\leq} \quad \mathcal{U}_t(\hat{d}_t) + \frac{1}{t+1}\nabla\mathcal{U}_t(\hat{d}_t)^\top(d_t^\star - \hat{d}_t) + \frac{1}{1+t}\epsilon_t + \frac{L}{(1+t)^2}
$$

$$
\overset{\text{convexity}}{\leq} \quad \mathcal{U}_t(\hat{d}_t) - \frac{1}{t+1}(\mathcal{U}_t(\hat{d}_t) - \mathcal{U}_t(\eta_t^\star)) + \frac{1}{1+t}\epsilon_t + \frac{L}{(1+t)^2} + \frac{1}{1+t}C_t
$$

$$
\mathcal{U}_t(\hat{d}_{t+1}) - \mathcal{U}_t(d_t^\star) \quad \leq \quad \mathcal{U}_t(\hat{d}_t) - \mathcal{U}_t(d_t^\star) - \frac{1}{t+1}(\mathcal{U}_t(\hat{d}_t) - \mathcal{U}_t(d_t^\star)) + \frac{1}{1+t}\epsilon_t + \frac{L}{(1+t)^2} + \frac{1}{1+t}C_t
$$

$$
\leq \quad \frac{t}{1+t}\left(\mathcal{U}_t(\hat{d}_t) - \mathcal{U}_t(d_t^\star)\right) + \frac{1}{1+t}\epsilon_t + \frac{L}{(1+t)^2} + \frac{1}{1+t}C_t
$$

$$
= \quad \frac{t}{1+t}\left(\mathcal{U}_t(\hat{d}_t) - \mathcal{U}_t(d_t^\star)\right) + \frac{1}{1+t}\epsilon_t + \frac{L}{(1+t)^2} + \frac{1}{1+t}C_t
$$

Now multiplying by $t+1$ both sides, and summing on $\frac{1}{T-1}\sum_{t=1}^{T-1}$. Using the shorthand $\rho_t(\hat{d}_t) = \mathcal{U}_t(\hat{d}_t) - \mathcal{U}_t(d_t^\star)$ we get:

$$
\frac{1}{T}\sum_{t=1}^{T-1}(t+1)\rho_{t+1}(\hat{d}_{t+1}) \leq \frac{1}{T}\sum_{t=1}^{T-1}t\rho_t(\hat{d}_t) + \frac{1}{T}\sum_{t=1}^{T-1}(\epsilon_t + C_t + L/(1+t))
$$

First notice that $\frac{1}{T-1}\sum_{t=1}^{T-1}\epsilon_t \leq \frac{B}{\sqrt{T}}\log(1/\delta)$ by Lemma in Mutný et al. [17] due to $\epsilon_t$ being martingale difference sequence. The other term is the sum on $\frac{1}{T-1}\sum_{t=1}C_t$ which appears in the main result. The sum on $\sum_{t=1}^{T-1}\frac{1}{1+t} \leq L\frac{\log T}{T}$. The rest is eliminated by the reccurence of the terms, and using that $\mathcal{U}(d|\mathcal{Z}_t, \{x_f^\star\}) \leq \mathcal{U}(d|\mathcal{Z}_{t-1}, \{x_f^\star\})$ for any $d$. This is due to set $\mathcal{Z}_t$ decreasing over time. We report the result in asymptotic notation as function of $T$ and $\log(1/\delta)$. $\qquad\square$

## D   Objective reformulation and linearization

For the main objective we try to optimize over a subset of $T$ trajectories $\mathbf{X} = \{\tau_i \in \mathcal{X}^H\}_{i=1}^T$. Let $\mathcal{X}^H$ be the set of sequences of inputs $\tau = (x_1, ..., x_H)$ where they consist of states in the search space $\mathcal{X}$. Furthermore, assume there exists, in the deterministic environment, a constraint such that $x_{h+1} \in \mathcal{C}(x_h)$ for all $h = 1, ..., H-1$. Then we seek to find the set $\mathbf{X}_*$, consisting of $T$ trajectories (possibly repeated), such that we solve the constrained optimization problem:

$$
\mathbf{X}_* = \arg\min_{\mathbf{X} \in \mathcal{X}^{TH}} \max_{z,z' \in \mathcal{Z}} \text{Var}[f(z) - f(z')|\mathbf{X}] \quad \text{s.t.} \quad x_{h+1} \in \mathcal{C}(x_h) \quad \forall t = 1, ..., h-1 \tag{23}
$$

We define the objective as:

$$
U(\mathbf{X}) = \max_{z,z' \in \mathcal{Z}} \text{Var}\left[f(z) - f(z')|\mathbf{X}\right] \tag{24}
$$

Our goal is to show that optimization over sequences can be simplified to state-action visitations as in Mutný et al. [17]. For this, we require that the objective depends additively involving terms $x, a$ separately. We formalize this in the next result. In order to prove the result, we utilize the theory of reproducing kernel Hilbert spaces [72].

**Lemma D.1** (Additivity of Best-arm Objective). *Let $\mathbf{X}$ be a collection of $t$ trajectories of length $H$. Assuming that $f \sim \mathcal{GP}(0, k)$. Assuming that $k$ has Mercer decomposition as $k(x, y) = \sum_k \lambda_k \phi_k(x)\phi_k(y)$.*

$$f(x) = \sum_k \phi_k(x)\theta_k \quad \theta_k \sim \mathcal{N}(0, \lambda_k).$$

*Let $d_{\mathbf{X}}$ be the visitation of the states-action in the trajectories in $\mathbf{X}$, as $d_{\mathbf{X}} = \frac{1}{TH}\sum_{t=1}^{T}\sum_{x,a\in\tau_t}\delta_{x,a}$, where the $\delta_{x,a}$ represent delta function supported on $x, a$. Then optimization of the objective Eq. (23) can be rewritten as:*

$$U(d_{\mathbf{X}}) = \frac{1}{TH}\max_{z,z'\in\mathcal{Z}}||\Phi(z) - \Phi(z')||^2_{\mathbf{V}(d_{\mathbf{X}})^{-1}},$$

*where $\mathbf{V}(d) = \sum_i\sum_{x,a\in\tau_i} d(x,a)\Phi(x)\Phi(x)^\top + \mathbf{I}\sigma^2/(TH)$ is a operator $\mathbf{V}(d) : \mathcal{H}_k \to \mathcal{H}_k$, the norm is RKHS norm, and $\Phi(z)_k = \phi_k(z)$.*

*Proof.* Notice that the posterior GP of any two points $z, z'$ is $(f(z), f(z')) = \mathcal{N}((\mu(z), \mu(z')), \mathbf{K}_{z,z'})$, where $\mathbf{K}_{z,z'}$ is posterior kernel (consult Rasmussen and Williams [28] for details) defined via a posterior kernel $k_{\mathbf{X}}(z, z') = k(z, z') - k(z, \mathbf{X})(\mathbf{K}(\mathbf{X}, \mathbf{X}) + \sigma^2\mathbf{I})^{-1}k(\mathbf{X}, z')$.

Utilizing $k(z, z') = \Phi(z)^\top\Phi(z)$ (RKHS inner product) with the Mercer decomposition we know that $k_t(z) = \Phi(\mathbf{X})\phi(z)$. Applying the matrix inversion lemma, the above can be written as using $\mathbf{V} = \sum_{t=1}^{T}\sum_{x\in\tau_t}\Phi(x)\Phi(x)^\top + \sigma^2\mathbf{I}_{\mathcal{H}}$.

$$
\begin{aligned}
k_{\mathbf{X}}(z, z') &= k(z, z') - k_t(z)^\top(\mathbf{K}_{\mathbf{X},\mathbf{X}} + \sigma^2\mathbf{I})^{-1}k_t(z') \\
&\overset{\text{Mercer}}{=} \Phi(z)^\top\Phi(z') - \Phi(z)^\top\Phi(\mathbf{X})^\top(\Phi(\mathbf{X})\Phi(\mathbf{X})^\top + \sigma^2\mathbf{I})^{-1}\Phi(\mathbf{X})\Phi(z') \\
&\overset{\text{Lemma D.3}}{=} \Phi(z)^\top\Phi(z') - \Phi(z)^\top\mathbf{V}^{-1}(\mathbf{V} - \mathbf{I}\sigma^2)\Phi(z') \\
&= \Phi(z)^\top\mathbf{V}^{-1}\mathbf{V}\Phi(z') - \Phi(z)^\top\mathbf{V}^{-1}(\mathbf{V} - \mathbf{I}\sigma^2)\Phi(z') \\
&= \Phi(z)^\top\mathbf{V}^{-1}(\mathbf{V} - \mathbf{V} + \mathbf{I}\sigma^2)\Phi(z')
\end{aligned}
$$

Leading finally to:

$$k_{\mathbf{X}}(z, z') = \sigma^2\Phi(z)^\top\left(\sum_{t=1}^{T}\sum_{x\in\tau_t}\Phi(x)\Phi(x)^\top + \sigma^2\mathbf{I}_{\mathcal{H}_k}\right)^{-1}\Phi(z'). \tag{25}$$

Let us calculate $\mathrm{Var}[f(z) - f(z')|\mathbf{X}]$. The variance does not depend on the mean. Hence,

$$
\begin{aligned}
&\mathrm{Var}\left[f(z) - f(z')|\mathbf{X}\right] \\
&= \mathrm{Var}(f(z)) - \mathrm{Var}(f(z')) - 2\mathrm{Cov}(f(z), f(z')) \\
&= k_{\mathbf{X}}(z, z) + k_{\mathbf{X}}(z', z') - 2k_{\mathbf{X}}(z, z') \\
&\overset{(25)}{=} (\Phi(z) - \Phi(z'))\left(\sum_{t=1}^{T}\sum_{x\in\tau_t}\Phi(x)\Phi(x)^\top + \sigma^2\mathbf{I}_{\mathcal{H}_k}\right)^{-1}(\Phi(z) - \Phi(z')) \\
&= (\Phi(z) - \Phi(z'))\left(\frac{TH}{TH}\sum_{t=1}^{T}\sum_{x\in\mathcal{X}}\#(x \in \tau_t)\Phi(x)\Phi(x)^\top + \sigma^2\mathbf{I}_{\mathcal{H}_k}\right)^{-1}(\Phi(z) - \Phi(z'))
\end{aligned}
$$

to arrive at:

$$\mathrm{Var}\left[f(z) - f(z')|\mathbf{X}\right] = \frac{(\Phi(z) - \Phi(z'))}{TH}\left(\sum_{x\in\mathcal{X}} d(\mathbf{X})\Phi(x)\Phi(x)^\top + \frac{\sigma^2}{TH}\mathbf{I}_{\mathcal{H}_k}\right)^{-1}(\Phi(z) - \Phi(z')) \tag{26}$$

The symbol $\#$ counts the number of occurrences. Notice that we have been able to show that the objective decomposes over state-action visitations as $d_{\mathbf{X}}$ decomposes over their visitations $\qquad\square$

Note that the objective equivalence does *not* imply that optimization problem in Eq. (23) is equivalent to finding,

$$d_* = \arg\min_{d_\pi \in \mathcal{D}} \max_{z,z' \in \mathcal{Z}} ||\Phi(z) - \Phi(z')||_{\mathbf{V}(d_\pi)^{-1}}. \tag{27}$$

In other words, optimization over trajectories and optimization over $d_\pi \in \mathcal{D}$ is not equivalent. The latter is merely a continuous relaxation of discrete optimization problems to the space of Markov policies. It is in line with the classical relaxation approach addressed in experiment design literature with a rich history, e.g., Chaloner and Verdinelli [73]. For introductory texts on the topic, consider Pukelsheim [74] for the statistical perspective and Boyd and Vandenberghe [75] for the optimization perspective. However, as Mutný et al. [17] points out, reducing the relaxed objective does reduce the objective as Eq. (23) as well. In other words, by optimizing the relaxation with a larger budget of trajectories or horizons, we are able to decrease Eq. (23) as well.

For completeness, we state the auxiliary lemma. We make use of the Sherman-Morrison-Woodbury (SMW) formula, [76]:

**Lemma D.2** (Sherman-Morrison-Woodbury (SMW)). *Let $\mathbf{A} \in \mathbb{R}^{n \times q}$ and $\mathbf{D} \in \mathbb{R}^{q \times q}$ then:*

$$(\mathbf{A}^\top \mathbf{D} \mathbf{A} + \rho^2 \mathbf{I})^{-1} = \rho^{-2} \mathbf{I} - \rho^{-2} A^T (\mathbf{D}^{-1} \rho^2 + \mathbf{A} \mathbf{A}^\top)^{-1} \mathbf{A}. \tag{28}$$

*Here we do the opposite, and invert an $n \times n$ matrix instead of a $q \times q$ one.*

to show the following.

**Lemma D.3** (Matrix Inversion Lemma). *Let $\mathbf{A} \in \mathbb{R}^{n \times q}$ then*

$$\mathbf{A}^\top (\mathbf{A} \mathbf{A}^\top + \rho^2 \mathbf{I})^{-1} = (\mathbf{A}^\top \mathbf{A} + \rho^2 \mathbf{I})^{-1} \mathbf{A}^\top. \tag{29}$$

*Note that instead of inverting $n \times n$ matrix, we can invert a $q \times q$ matrix.*

*Proof.*

$$
\begin{aligned}
\mathbf{A}^\top (\mathbf{A} \mathbf{A}^\top + \rho^2 \mathbf{I})^{-1} &\overset{\text{SMW}}{=} \mathbf{A}^\top (\rho^{-2} \mathbf{I} - \rho^{-2} \mathbf{A}(\rho^2 \mathbf{I} + \mathbf{A}^\top \mathbf{A})^{-1} \mathbf{A}^\top) \\
&= (\rho^{-2} \mathbf{I} - \rho^{-2} \mathbf{A}^\top \mathbf{A}(\rho^2 \mathbf{I} + \mathbf{A}^\top \mathbf{A})^{-1}) \mathbf{A}^\top \\
&= (\rho^{-2}(\rho^2 \mathbf{I} + \mathbf{A}^\top \mathbf{A}) - \rho^{-2} \mathbf{A}^\top \mathbf{A})(\rho^2 \mathbf{I} + \mathbf{A}^\top \mathbf{A})^{-1} \mathbf{A}^\top \\
&= (\mathbf{A}^\top \mathbf{A} + \rho^2 \mathbf{I})^{-1} \mathbf{A}^\top
\end{aligned}
$$

$\square$

## D.1 Objective formulation for general kernel methods

The previous discussion also allows us to write the objective in terms of the general kernel matrix instead of relying on finite dimensional embeddings. The modification is very similar and relies again on Sherman-Mirrison-Woodbury lemma.

We now work backwards from (26), and first write the objective in terms of features of arbitrarily large size. Using the shorthand, $\tilde{\sigma}^2 = \sigma^2/TH$, let us define a diagonal matrix that describes the state-action distribution $\mathbf{D} = \text{diag}(\{d_x : x \in \mathcal{X}\})$ of the size $|\mathcal{X}| \times |\mathcal{X}|$, and $\Phi(\mathcal{X})$ which corresponds to the unique (possibly infinite-dimensional) embeddings of each element in $\mathcal{X}$, ordered in the same way as $\mathbf{D}$.

$$
\tilde{\sigma}^2 \left( \sum_{x \in \mathcal{X}} d(x) \Phi(x) \Phi(x) + \mathbf{I} \tilde{\sigma}^2 \right)^{-1} = \tilde{\sigma}^2 \left( \Phi(\mathcal{X})^T \mathbf{D} \Phi(\mathcal{X}) + \mathbf{I} \tilde{\sigma}^2 \right)^{-1}
$$

$$
= \mathbf{I} - \Phi(\mathcal{X})^T (\tilde{\sigma}^2 \mathbf{D}^{-1} + \Phi(\mathcal{X}) \Phi(\mathcal{X})^\top)^{-1} \Phi(\mathcal{X})
$$

If we then pre-multiply by $\Phi(z)^\top$ and $\Phi(z')$ we obtain:

$$
k_{\mathbf{X}}(z, z') = \Phi(z)^\top \Phi(z') - \Phi(z)^\top \Phi(\mathcal{X})^\top (\tilde{\sigma}^2 \mathbf{D}^{-1} + \Phi(\mathcal{X}) \Phi(\mathcal{X})^\top)^{-1} \Phi(\mathcal{X}) \Phi(z')
$$

Finally giving:

$$
k_{\mathbf{X}}(z, z') = k(z, z') - k(z, \mathcal{X})(\tilde{\sigma}^2 \mathbf{D}^{-1} + k(\mathcal{X}, \mathcal{X}))^{-1} k(\mathcal{X}, z') \tag{30}
$$

which allows us to calculate the objective for general kernel methods at the cost of an $|\mathcal{X}| \times |\mathcal{X}|$ inversion. Upon identifying the $z, z'$ that maximize the above, we can use them in an optimization procedure. This holds irrespective of whether the state space is discrete or continuous. In continuous settings however, we again require a parametrization of the infinite dimensional probability distribution by some finite means such as claiming that $\mathbf{D}_\theta$ contains some finite dimensional simplicity. This is what we do with the linear system example in Section 4.2.

## D.2 Linearizing the objective

To apply our method, we find ourselves having to frequently solve RL sub-problems where we try to maximize $\sum_{x,a} d(x,a)\nabla F(x,a)$. To approximately solve this problem in higher dimensions, it becomes very important to understand what the linearized functional looks like.

*Remark* D.4. Assume the same black-box model as in Lemma D.1, and further assume that we have a mixture of policies $\pi_{\text{mix}}$ with density $d_{\pi_{\text{mix}}}$, such that there exists a set $\mathbf{X}_{\text{mix}}$ satisfying $d_{\pi_{\text{mix}}} = \frac{1}{N}\sum_{x\in\mathbf{X}_{\text{mix}}} \delta_x$ for some integer $N$. Then:

$$\nabla F(d_{\pi_{\text{mix}}})(x,a) \propto -\left(\text{Cov}[f(z_*), f(x)|\mathbf{X}_{\text{mix}}] - \text{Cov}[f(z'_*), f(x)|\mathbf{X}_{\text{mix}}]\right)^2$$

where $z_*, z'_* = \arg\max_{z,z'\in\mathcal{Z}} \text{Var}[f(z) - f(z')]$.

*Proof.* To show this, we begin by defining:

$$\Sigma_{\theta,d} = \left(\sum_{x\in\mathcal{X}} \Phi(x)\Phi(x)^T d(x) + \sigma^2 I\right)^{-1}$$

$$z_*, z'_* = \arg\max_{z,z'\in\mathcal{Z}} ||\Phi(z) - \Phi(z')||^2_{\Sigma_{\theta,d}}$$

$$\tilde{z}_* = \Phi(z_*) - \Phi(z'_*)$$

In the definition above we dropped the constant pre-factors since they do not influence the maximizer of the gradient as they are related by a constant multiplicative factor.

It then follows, by applying Danskin's Theorem that:

$$\begin{aligned}
\nabla U(d)(x) &= \nabla \tilde{z}_*^T \Sigma_{\theta,d} \tilde{z}_* \\
&= \nabla\text{Tr}\left\{\tilde{z}_*\tilde{z}_*^T \Sigma_{\theta,d}\right\} \\
&= \text{Tr}\left\{\tilde{z}_*\tilde{z}_*^T \nabla\Sigma_{\theta,d}\right\} \\
&= -\text{Tr}\left\{\tilde{z}_*\tilde{z}_*^T \Sigma_{\theta,d}\Phi(x)\Phi(x)^T\Sigma_{\theta,d}\right\} \qquad (\text{as } \partial K^{-1} = -K^{-1}(\partial K)K^{-1}) \\
&= -\text{Tr}\left\{\tilde{z}_*^T \Sigma_{\theta,d}\Phi(x)\Phi(x)^T\Sigma_{\theta,d}\tilde{z}_*\right\} \\
&= -\left(\tilde{z}_*^T \Sigma_{\theta,d}\Phi(x)\right)\left(\Phi(x)^T\Sigma_{\theta,d}\tilde{z}_*\right) \\
&\propto -\left(\text{Cov}[f(z_*), f(x)] - \text{Cov}[f(z'_*), f(x)]\right)\left(\text{Cov}[f(x), f(z_*)] - \text{Cov}[f(x), f(z'_*)]\right) \\
&= -\left(\text{Cov}[f(z_*), f(x)] - \text{Cov}[f(z'_*), f(x)]\right)^2
\end{aligned}$$

$\square$

# E  Implementation details and Ablation study

In this section we provide implementation details, and show some studies into the effects of specific hyper-parameters. We note that the implementation code will be made public after public review.

## E.1  Approximating the set of maximizers using Batch BayesOpt

We give details of the two methods used for approximating the set of potential maximizers. In particular, we first focus on Thompson Sampling [58]:

$$\mathcal{Z}_{cont}^{(TS)} = \left\{\arg\max_{x\in\mathcal{X}_c} f_i(x) : f_i \sim \mathcal{GP}(\mu_t, \sigma_t)\right\}_{i=1}^K$$

where $K$ is a new hyper-parameter influencing the accuracy of the approximation of $\mathcal{Z}$. We found that the algorithm could be too exploratory in certain scenarios. Therefore, we also propose an alternative that encourages exploitation by guiding the maximization set using BayesOpt through the UCB acquisition function [71]:

$$\mathcal{Z}_{cont}^{(UCB)} = \left\{\arg\max_{x\in\mathcal{X}_c} \mu_t(x) + \beta_i\sigma_t(x) : \beta_i \in \mathcal{B}\right\}_{i=1}^K$$

where $\mathcal{B} = \texttt{linspace(0, 2.5, K)}$ which serves as scaling for the size of set $\mathcal{Z}_{cont}^{(UCB)}$. Both cases reduce optimization over $\mathcal{Z}$ to enumeration as with discrete cases.

## E.2 Benchmark Details

For all benchmarks, aside from the knorr pyrazole synthesis example, we use a standard squared exponential kernel for the surrogate Gaussian Process:

$$k_{rbf}(x, x') = \sigma_{rbf}^2 \exp\left(-\frac{||x - x'||_2^2}{2\ell_{rbf}}\right)$$

where $\sigma_{rbf}^2$ is the prior variance of the kernel, and $\ell_{rbf}$ the kernel. We fix the values of all the hyper-parameters a priori and use the same for all algorithms. The hyper-parameters for each benchmarks are included in Table 2.

For the knorr pyrazole synthesis example, we further set $\alpha_{ode} = 0.6$, $\alpha_{rbf} = 0.001$, $k_1 = 10$, $k_2 = 874$, $k_3 = 19200$, $\alpha_{sig} = 5$. Recall we are using a finite dimensional estimate of a GP such that:

$$f(x) \approx \omega_{ode}\Phi_{ode}(x) + \sum_{i=1}^{M} \omega_{rbf,i}\Phi_{rbf}(x) \tag{31}$$

in this case we set a prior to the ODE weight such that $\omega_{ode} \sim \mathcal{N}(0.6, 0.0225)$. This is incorporating two key pieces of prior knowledge that (a) the product concentration should be positive, and (b) we expect a maximum product concentration between 0.15 and 0.45.

The number of features for each experiment, $M$, is set to be $M = |\mathcal{X}|$ in the discrete cases and $M = \min\left(2^{5+d}, 512\right)$ where $d$ is the problem dimensionality.

In the case of Local Search Region BayesOpt (LSR) [22] we set the exploration hyper-parameter to be $\gamma = 0.01$ in all benchmarks.

Table 2: Benchmark and hyper-parameter information. $\Delta_{max}$ represents the size of the box constraints in the traditional benchmarks. For the synchronous benchmarks and for SnAr we used a noise level of $\sigma^2 = 0.001$. For the asynchronous benchmarks, and the knorr pyrazole example we used $\sigma^2 = 0.0001$. For the Ypacarai example we used $\sigma^2 = 0.001$ and $\sigma^2 = 0.01$ for the episodic and immediate feedback respectively.

| Benchmark Name | $\Delta_{max}$ | Variance $\sigma_{rbf}$ | Lengthscale $\ell_{rbf}$ |
|---|---|---|---|
| Knorr pyrazole | – | 0.001 | 0.1 |
| Constrained Ypacarai | – | 1 | 0.2 |
| Branin2D | 0.05 | 0.6 | 0.15 |
| Hartmann3D | 0.1 | 2.0 | 0.13849 |
| Hartmann6D | 0.2 | 1.7 | 0.22 |
| Michalewicz2D | 0.05 | 0.35 | 0.179485 |
| Michalewicz3D | 0.1 | 0.85 | 0.179485 |
| Levy4D | 0.1 | 0.6 | 0.14175 |
| SnAr | 0.1 | 0.8 | 0.2 |

## E.3 Ypacarai Lake

Samaniego et al. [20] investigated the use of Bayesian Optimization for monitoring the lake quality in Lake Ypacarai in Paraguay. We extend the benchmark to include additional transition constraints, as well as initial and end-point constraints. These are all shown in Figure 12.

## E.4 Free-electron Laser

We use the simulator from Mutný et al. [67] that optimizes quadrupole magnet orientations for our experiment with varying noise levels. We use a 2-dimensional variant of the simulator. We discretize the system on $10 \times 10$ grid and assume that the planning horizon $H = 100$. The simulator itself is a GP fit with $\gamma = 0.4$, hence we use this value. Then we make a choice that the noise variance is proportional to the change made as $\sigma^2(x, a) = s(1 + w||x - a||^2)$, where $s = 0.01$ and $w = 20$. Note that $x \in [-0.5, 0.5]^2$ in this modeling setup. This means that local steps are indeed very desired. We showcase the difference to classical BayesOpt, which uses the worst-case variance $\sigma = \sup_{x,a} s(1 + w||x - a||^2)$ for modeling as it does not take into account the state in which the system is. We see that the absence of state modeling leads to a dramatic decrease in performance as indicated by much higher inference regret in Figure 3b.

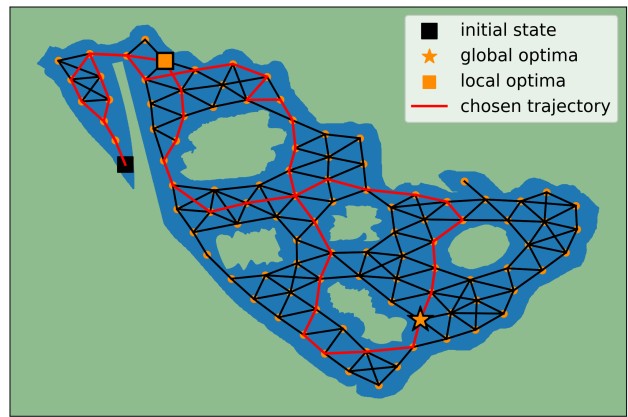

Figure 12: Lake Ypacari with the added movement constraints. We show one local optimum and one global one. The constraints of the problem requiring beginning and ending the optimization in the dark square.

## E.5 Ablation Study

### E.5.1 Number of mixture components

We investigate the effect number of components used in the mixture policy when optimizing the Frank-Wolfe algorithm. We tested on the four real-world problem using $N = 1, 10$ and $25$. In the Ypacari example (see Figure 14) we see very little difference in the results, while in the Knorr pyrazole synthesis (see Figure 13) we observe a much bigger difference. A single component gives a much stronger performance than multiple ones – we conjecture this is because the optimum is on the edge of the search space, and adding more components makes the policy stochastic and less likely to reach the boarder (given episodes are of length ten and ten right-steps are required to reach the boarder).

Overall, it seems the performance of a single component is better or at worst comparable as using multiple components. This is most likely due to the fact that we only follow the Markovian policies for a single time-step before recalculating, making the overall impact of mixture policies smaller. Based on this, we only present the single-component variant in the main paper.

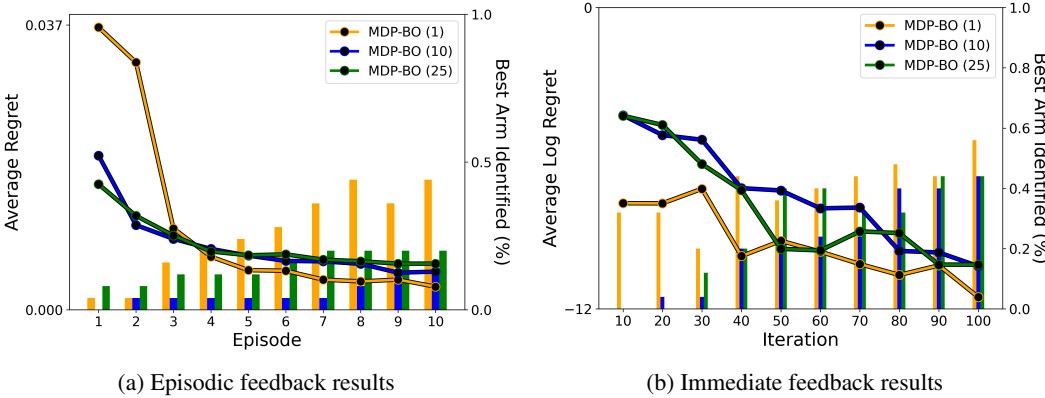

(a) Episodic feedback results

(b) Immediate feedback results

Figure 13: Ablation study on the number of mixture components on the Knorr pyrazole synthesis benchmark

### E.5.2 Size of batch for approximating the set of maximizers

We explore the effect of the number of maximizers, $K$, in the maximization sets $\mathcal{Z}_{cont}^{(TS)}$ and $\mathcal{Z}_{cont}^{(UCB)}$. Overall we found the performance of the algorithm to be fairly robust to the size of the set in all benchmarks, with a higher $K$ generally leading to a little less spread in the performance.

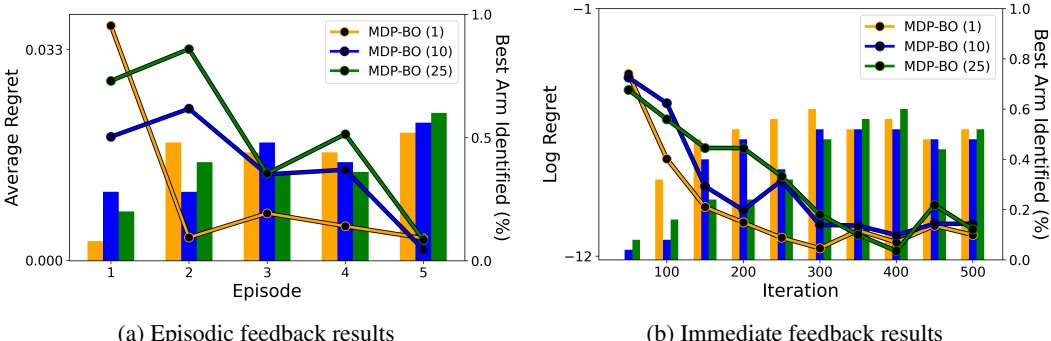

(a) Episodic feedback results      (b) Immediate feedback results

Figure 14: Ablation study on the number of mixture components on the Ypacarai benchmark

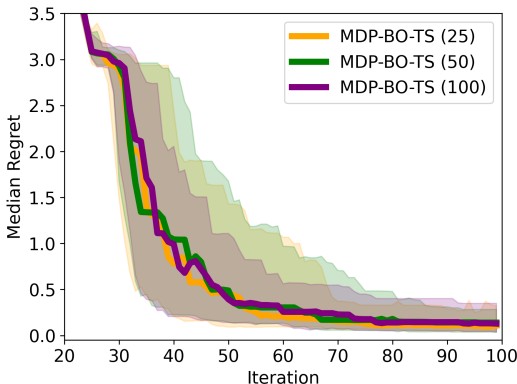

Figure 15: Ablation study into the size of the Thompson Sampling maximization set in the asynchronous Hartmann3D function. We can see that the performance of the algorithm is very similar for all values of $K = 25, 50, 100$.

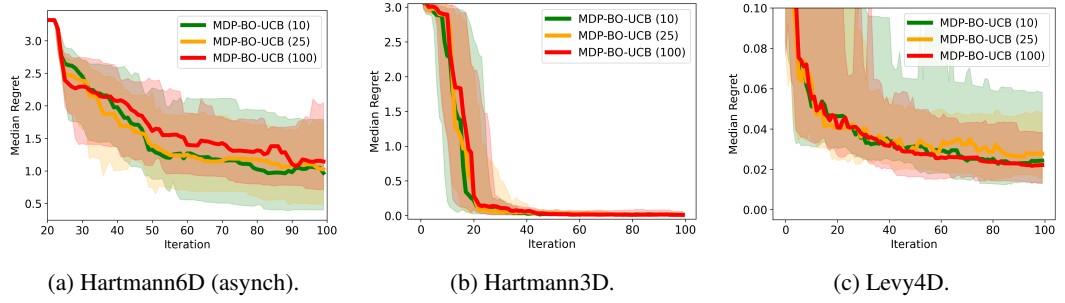

(a) Hartmann6D (asynch).      (b) Hartmann3D.      (c) Levy4D.

Figure 16: Ablation study into the size of the UCB maximization set in a variety of benchmarks. We can see that the performance of the algorithm is very similar for all values of $K = 10, 25, 100$.

## F    $\mathcal{XY}$-allocation vs $\mathcal{G}$-allocation

Our objective is motivated by hypothesis testing between different arms (options) $z$ and $z'$. In particular,

$$U(d) = \max_{z', z \in \mathcal{Z}} \text{Var}[f(z) - f(z')|d_{\mathbf{X}}]. \tag{32}$$

One could maximize the information of the location of the optimum, as it has a Bayesian interpretation. This is at odds in frequentist setting, where such interpretation does not exists. Optimization of information about the maximum has been explored before, in particular via information-theoretic acquisition functions [29, 77–79]. However, good results (in terms of regret) have been achieved by focusing only on yet another surrogate to this, namely, the value of the maximum [80]. This is chiefly due to problem of dealing with the distribution of $f(x^\star)$. Defining a posterior value for $f(z)$ is easy.

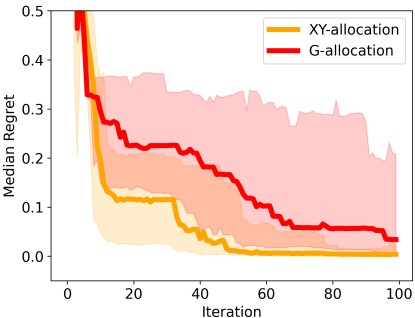

Figure 17: Comparison of using $\mathcal{XY}$-allocation against $\mathcal{G}$-allocation as the basis for the objective. In both cases the maximization sets were created using Thompson Sampling. Overall the performances were often similar, however in a few examples, such as Branin2D which we showcase here, $\mathcal{G}$-allocation performed very poorly. This is consistent with what we can expect from the bandits literature.

Using, this and the worst-case perspective, an alternative way to approximate the best-arm objective, could be:

$$\tilde{U}(d) = \max_{z \in \mathcal{Z}} \text{Var}[f(z)|\mathbf{X}_d]. \tag{33}$$

What are we losing by not considering the differences? The original objective corresponds to the $\mathcal{XY}$-allocation in the bandits literature. The modified objective will, in turn, correspond to the $\mathcal{G}$-allocation, which has been argued can perform arbitrarily worse as it does not consider the differences, e.g. see Appendix A in Soare et al. [46]. We nonetheless implemented the algorithm with objective (33), and found the results to be as expected: performance was very similar *in general*, however in some cases not considering the differences led to much poorer performance. As an example, see Figure 17 for results on the synchronous Branin2D benchmark.

## G   Practical Planning for Continuous MDPs

From remark D.4 it becomes clear that for decreasing covariance functions, such as the squared exponential, $\nabla F$ will consist of two modes around $z_*$ and $z'_*$ The sub-problem seems to find a sequence that maximizes the *sum* of gradients, therefore the optimal solution will try to reach one of the two modes as quickly as possible. For shorter time horizons, the path will reach whichever mode is closest, and for large enough horizons, the sum will be maximized by reaching the larger of the two modes.

Therefore we can approximately solve the problem by checking the value of the sub-problem objective in (13) for the shortest paths from $x_{t-1} \to z_*$ and $x_{t-1} \to z'_*$, which are trivial to find under the constraints in (13). Note that the paths might not necessarily be optimal, as they may be improved by small perturbations, e.g., there might be a small deviation that allows us to visit the smaller mode on the way to the larger mode increasing the overall value of the sum of gradients, however, they give us a good and quick approximation.

## H   Kernel for ODE Knorr pyrazole synthesis

The kernel is based on the following ODE model, which is well known in the chemistry literature and given in [27].

$$R_1 = k_1 y_2 y_3 - k_2 y_4 y_5 \tag{34}$$
$$R_2 = k_3 y_4 \tag{35}$$

and then:

$$\frac{dy_1}{dt} = R_2$$

$$\frac{dy_2}{dt} = -R_1$$

$$\frac{dy_3}{dt} = -R_1$$

$$\frac{dy_4}{dt} = R_1 - R_2$$

$$\frac{dy_5}{dt} = R_1 + R_2$$

Our main goal is to optimize the product concentration of the reaction, which is given by $y_1$. We do this by sequentially querying the reaction, where we select the residence time, and the initial conditions of the ODE, in the form $y_0 = [0, A, B, 0, 0]$, where $A = 1 - B$.

Due to the non-linearity in Eq. (34) we are unable to fit a GP to the process directly. Instead, we first linearize the ODE around two equilibrium points. The set of points of equilibrium are given by:

$$S_1 = \{y_1 = a_1, y_2 = b_1, y_3 = 0, y_4 = 0, y_5 = c_1 | a_1, b_1, c_1 \in \mathbf{R}\}$$
$$S_2 = \{y_1 = a_2, y_2 = 0, y_3 = b_2, y_4 = 0, y_5 = c_2 | a_1, b_1, c_1 \in \mathbf{R}\}$$

And the Jacobian of the system is:

$$\mathbf{J} = \begin{bmatrix} 0 & 0 & 0 & k_3 & 0 \\ 0 & -k_1 y_3 & -k_1 y_2 & k_2 y_5 & k_2 y_4 \\ 0 & -k_1 y_3 & -k_1 y_2 & k_2 y_5 & k_2 y_4 \\ 0 & k_1 y_3 & k_1 y_2 & -k_2 y_5 - k_3 & -k_2 y_4 \\ 0 & k_1 y_3 & k_1 y_2 & -k_2 y_5 + k_3 & -k_2 y_4 \end{bmatrix}$$

Giving:

$$\mathbf{J}_1 = \mathbf{J}|_{S_1} = \begin{bmatrix} 0 & 0 & 0 & k_3 & 0 \\ 0 & 0 & -k_1 b_1 & k_2 c_1 & 0 \\ 0 & 0 & -k_1 b_1 & k_2 c_1 & 0 \\ 0 & 0 & k_1 b_1 & -k_2 c_1 - k_3 & 0 \\ 0 & 0 & k_1 b_1 & -k_2 c_1 + k_3 & 0 \end{bmatrix}$$

$$\mathbf{J}_2 = \mathbf{J}|_{S_2} = \begin{bmatrix} 0 & 0 & 0 & k_3 & 0 \\ 0 & -k_1 b_2 & 0 & k_2 c_2 & 0 \\ 0 & -k_1 b_2 & 0 & k_2 c_2 & 0 \\ 0 & k_1 b_2 & 0 & -k_2 c_2 - k_3 & 0 \\ 0 & k_1 b_2 & 0 & -k_2 c_2 + k_3 & 0 \end{bmatrix}$$

Unfortunately, since the matrices are singular, we do not get theoretical results on the quality of the linearization. However, linearization is still possible, with the linear systems given by:

$$\frac{d\vec{y}}{dt} = \mathbf{J}_1 \vec{y} \qquad \frac{d\vec{y}}{dt} = \mathbf{J}_2 \vec{y}$$

We focus on the first system for now. The matrix has the following eigenvalues:

$$\lambda_{1,2} = -\frac{1}{2}\left(b_1 k_1 + c_1 k_2 + k_3 \pm \sqrt{b_1^2 k_1^2 + c_1^2 k_2^2 + k_3^2 + 2b_1 c_1 k_1 k_2 - 2k_3(b_1 k_1 - c_1 k_2)}\right)$$

$$\lambda_{3,4,5} = 0$$

Note that the three eigenvalues give us the corresponding solution based on their (linearly separable) eigenvectors:

$$v_3 = \begin{bmatrix} 1 & 0 & 0 & 0 & 0 \end{bmatrix}, \quad v_4 = \begin{bmatrix} 0 & 1 & 0 & 0 & 0 \end{bmatrix}, \quad v_5 = \begin{bmatrix} 0 & 0 & 0 & 0 & 1 \end{bmatrix}$$

$$\vec{y}(t) = p_3 v_3 + p_4 v_4 + p_5 v_5$$

where $p_i$ are constants. The behaviour of the ODE when this is not the case will depend on whether the remaining eigenvalues will be real or not. However, note:

$$b_1^2 k_1^2 + c_1^2 k_2^2 + k_3^2 + 2b_1 c_1 k_1 k_2 - 2k_3(b_1 k_1 - c_1 k_2) \geq b_1^2 k_1^2 + k_3^2 - 2b_1 k_1 k_3 = (b_1 k_1 - k_3)^2 \geq 0$$

and therefore all eigenvalues will always be real. Therefore we can write down the solution as:

$$\vec{y}(t) = p_1 v_1 e^{\lambda_1 t} + p_2 v_2 e^{\lambda_2 t} + p_3 v_3 + p_4 v_4 + p_5 v_5$$

where we ignore the case of repeated eigenvalues for simplicity (this is the case where $b_1^2 k_1^2 + 2b_1 c_1 k_1 k_2 + c_1^2 k_2^2 - 2k_3(b_1 k_1 - c_1 k_2) + k_3^2$ is exactly equal to zero). We further note that the eigenvalues will be non-negative as:

$$b_1 k_1 + c_1 k_2 + k_3 = \sqrt{(bk_1 + c_1 k_2 + k_3)^2}$$

$$= \sqrt{b_1^2 k_1^2 + c^2 k_2^2 + k_3^2 + 2b_1 c_1 k_1 k_2 + 2c_1 k_2 k_2 + 2b_1 k_1 k_3}$$

$$\geq \sqrt{b_1^2 k_1^2 + c_1^2 k_2^2 + k_3^2 + 2b_1 c_1 k_1 k_2 + 2c_1 k_2 k_2 - 2b_1 k_1 k_3}$$

therefore:

$$\lambda_1 \leq \lambda_2 \leq 0$$

which means the solutions will always be a linear combination of exponentially decaying functions of time plus constants.

The eigenvectors have the closed form:

$$v_1 = \begin{pmatrix} 1, \\ \frac{1}{2}\left(b_1 k_1 + c_1 k_2 - k_3 + \sqrt{b_1^2 k_1^2 + 2b_1 c_1 k_1 k_2 + c_1^2 k_2^2 - 2(b_1 k_1 - c_1 k_2)k_3 + k_3^2}\right)/k_3, \\ \frac{1}{2}\left(b_1 k_1 + c_1 k_2 - k_3 + \sqrt{b_1^2 k_1^2 + 2b_1 c_1 k_1 k_2 + c_1^2 k_2^2 - 2(b_1 k_1 - c_1 k_2)k_3 + k_3^2}\right)/k_3, \\ -\frac{1}{2}\left(b_1 k_1 + c_1 k_2 + k_3 + \sqrt{b_1^2 k_1^2 + 2b_1 c_1 k_1 k_2 + c_1^2 k_2^2 - 2(b_1 k_1 - c_1 k_2)k_3 + k_3^2}\right)/k_3, \\ -\frac{1}{2}\left(b_1 k_1 + c_1 k_2 - 3k_3 + \sqrt{b_1^2 k_1^2 + 2b_1 c_1 k_1 k_2 + c_1^2 k_2^2 - 2(b_1 k_1 - c_1 k_2)k_3 + k_3^2}\right)/k_3 \end{pmatrix} = \begin{pmatrix} 1, \\ -\lambda_1/k_3 - 1, \\ -\lambda_1/k_3 - 1, \\ \lambda_1/k_3, \\ \lambda_1/k_3 + 2 \end{pmatrix}$$

$$v_2 = \begin{pmatrix} 1, \\ \frac{1}{2}\left(b_1 k_1 + c_1 k_2 - k_3 - \sqrt{b_1^2 k_1^2 + 2b_1 c_1 k_1 k_2 + c_1^2 k_2^2 - 2(b_1 k_1 - c_1 k_2)k_3 + k_3^2}\right)/k_3, \\ \frac{1}{2}\left(b_1 k_1 + c_1 k_2 - k_3 - \sqrt{b_1^2 k_1^2 + 2b_1 c_1 k_1 k_2 + c_1^2 k_2^2 - 2(b_1 k_1 - c_1 k_2)k_3 + k_3^2}\right)/k_3, \\ -\frac{1}{2}\left(b_1 k_1 + c_1 k_2 + k_3 - \sqrt{b_1^2 k_1^2 + 2b_1 c_1 k_1 k_2 + c_1^2 k_2^2 - 2(b_1 k_1 - c_1 k_2)k_3 + k_3^2}\right)/k_3, \\ -\frac{1}{2}\left(b_1 k_1 + c_1 k_2 - 3k_3 - \sqrt{b_1^2 k_1^2 + 2b_1 c_1 k_1 k_2 + c_1^2 k_2^2 - 2(b_1 k_1 - c_1 k_2)k_3 + k_3^2}\right)/k_3 \end{pmatrix} = \begin{pmatrix} 1, \\ -\lambda_2/k_3 - 1, \\ -\lambda_2/k_3 - 1, \\ \lambda_2/k_3, \\ \lambda_2/k_3 + 2 \end{pmatrix}$$

We are optimizing over initial set of conditions $y_0 = [0, A, B, 0, 0]$, so solving for the specific values of the constants gives:

$$p_1 = \frac{\lambda_2}{\lambda_1 - \lambda_2} B$$

$$p_2 = -\frac{\lambda_1}{\lambda_1 - \lambda_2} B$$

$$p_3 = B$$

$$p_4 = A - B$$

$$p_5 = 2B$$

Finally, since we are setting $A = 1 - B$ and we are only optimizing the first component of $\vec{y}$ we can obtain it in closed form:

$$y_1(t, B) = \frac{\lambda_2}{\lambda_1 - \lambda_2} B e^{\lambda_1 t} - \frac{\lambda_1}{\lambda_1 - \lambda_2} B e^{\lambda_2 t} + B$$

$$= B\left(\frac{\lambda_2}{\lambda_1 - \lambda_2} e^{\lambda_1 t} - \frac{\lambda_1}{\lambda_1 - \lambda_2} e^{\lambda_2 t} + 1\right) \tag{36}$$

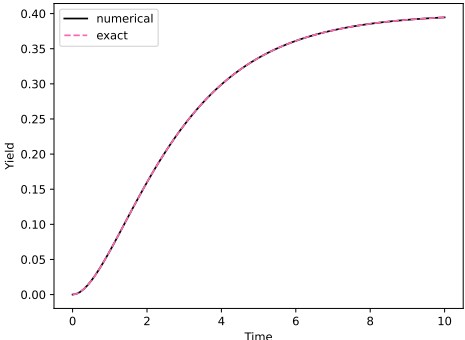

Figure 18: Comparing the numerical solution against the solutions found in equation (36).

The second ODE is very similar to the first, recall it depends has the following matrix:

$$\mathbf{J}_2 = \mathbf{J}|_{S_2} = \begin{bmatrix} 0 & 0 & 0 & k_3 & 0 \\ 0 & -k_1 b_2 & 0 & k_2 c_2 & 0 \\ 0 & -k_1 b_2 & 0 & k_2 c_2 & 0 \\ 0 & k_1 b_2 & 0 & -k_2 c_2 - k_3 & 0 \\ 0 & k_1 b_2 & 0 & -k_2 c_2 + k_3 & 0 \end{bmatrix}$$

The resulting ODE is symmetric to the alternate linearization giving the same solution:

$$y(t) = p_1 v_1 e^{\lambda_1 t} + p_2 v_2 e^{\lambda_2 t} + p_3 v_3 + p_4 v_4 + p_5 v_5$$

with the only difference being the eigenvectors now are:

$$v_3 = \begin{bmatrix} 1 & 0 & 0 & 0 & 0 \end{bmatrix}, \quad v_4 = \begin{bmatrix} 0 & 0 & 1 & 0 & 0 \end{bmatrix}, \quad v_5 = \begin{bmatrix} 0 & 0 & 0 & 0 & 1 \end{bmatrix}$$

which in turn leads to solutions of the form:

$$
\begin{aligned}
y_1(t, B) &= \frac{\lambda_2}{\lambda_1 - \lambda_2} A e^{\lambda_1 t} - \frac{\lambda_1}{\lambda_1 - \lambda_2} A e^{\lambda_2 t} + A \\
&= A \left( \frac{\lambda_2}{\lambda_1 - \lambda_2} e^{\lambda_1 t} - \frac{\lambda_1}{\lambda_1 - \lambda_2} e^{\lambda_2 t} + 1 \right)
\end{aligned}
$$

where $A = 1 - B$. Note that we now have four different eigenvalues, which depend on the linearization points:

$$\lambda_{1,2}^{(1)} = -\frac{1}{2} \left( b_1 k_1 + c_1 k_2 + k_3 \pm \sqrt{b_1^2 k_1^2 + c_1^2 k_2^2 + k_3^2 + 2 b_1 c_1 k_1 k_2 - 2 k_3 (b_1 k_1 - c_1 k_2)} \right)$$

$$\lambda_{1,2}^{(2)} = -\frac{1}{2} \left( b_2 k_1 + c_2 k_2 + k_3 \pm \sqrt{b_2^2 k_1^2 + c_2^2 k_2^2 + k_3^2 + 2 b_2 c_2 k_1 k_2 - 2 k_3 (b_2 k_1 - c_2 k_2)} \right)$$

Giving solutions:

$$y_1^{(1)}(t, B) = B \left( \frac{\lambda_2^{(1)}}{\lambda_1^{(1)} - \lambda_2^{(1)}} e^{\lambda_1^{(1)} t} - \frac{\lambda_1^{(1)}}{\lambda_1^{(1)} - \lambda_2^{(1)}} e^{\lambda_2^{(1)} t} + 1 \right)$$

$$y_1^{(2)}(t, B) = A \left( \frac{\lambda_2^{(2)}}{\lambda_1^{(2)} - \lambda_2^{(2)}} e^{\lambda_1^{(2)} t} - \frac{\lambda_1^{(2)}}{\lambda_1^{(2)} - \lambda_2^{(2)}} e^{\lambda_2^{(2)} t} + 1 \right)$$

Due to the length of the derivation, we confirm that our analysis is correct by comparing the numerical solution of the ODE to the exact solution we found in Figure 18. Finally, we look at interpolating between the two solutions; so given the solutions $y^{(1)}(t, B)$ and $y^{(2)}(t, B)$ corresponding to the linearization with stationary point in $S_1$ and $S_2$ respectively, we consider a solution of the form:

$$y(t, B | k_1, k_2, k_3, \alpha) = (1 - \mathcal{S}(B)) y^{(1)}(t, B) + \mathcal{S}(B) y^{(2)}(t, B) \tag{37}$$

where $\mathcal{S}(x) := (1 + e^{-\alpha_{sig}(x-0.5)})^{-1}$ is a sigmoid function centered at $B = 0.5$ and where we have introduced a new hyper-parameter $\alpha_{sig}$. Finally, given Eq. (37) we can obtain the kernel. In particular, we want (37) to be a feature we are predicting on; therefore the kernel is simply the (dot) product of the features therefore:

$$k_{ode}((t, B), (t', B')) = y(t, B | k_1, k_2, k_3, \alpha) \times y(t', B' | k_1, k_2, k_3, \alpha)$$

And because we know we are simply approximating the data we can simply correct the model by adding an Gaussian Process correction; giving us the final kernel:

$$k_{joint}((t, B), (t', B')) = \alpha_{ode} k_{ode}((t, B), (t', B')) + \alpha_{rbf} k_{rbf}((t, B), (t', B'))$$

where $\alpha_{ode}$ and $\alpha_{rbf}$ are parameters we can learn, e.g. using the marginal likelihood.

