# OpenReview forum: "Transition Constrained Bayesian Optimization via Markov Decision Processes"
_NeurIPS.cc/2024/Conference — NeurIPS 2024 poster_

### Official Review · Reviewer_2XmZ · 2024-07-11

**Soundness:** 3
**Presentation:** 3
**Contribution:** 3
**Rating:** 6
**Confidence:** 3

**Summary:**

This paper explores extending Bayesian optimization to incorporate transition constraints through state dynamics. The approach treats the problem as an optimal control problem. For planning purposes, known state dynamics model the transition constraints, while the unknown objective is represented using a Gaussian process. To address this, the paper introduces a tractable acquisition function—an upper bound on the maximum probability of selecting the wrong optimizer. This acquisition function serves as a cost metric in a planning scenario, which is solved using the Frank-Wolfe algorithm. The paper’s effectiveness is evaluated across various benchmark tasks.

**Strengths:**

The paper presents a framework for optimizing challenging-to-evaluate black-box functions with transition constraints, which could have practical applications. Its overall structure is well-organized.

The proposed solution strategy for solving the planning problem using the Frank-Wolfe algorithm is novel and represents a valuable contribution to the research community. Additionally, considering both discrete and continuous state problems is commendable.

While the experiments are fair, including a theoretical synthetic example would have further strengthened the paper.

**Weaknesses:**

General Presentation:

Generally, I believe the paper’s presentation could be enhanced. Specifically, I am not very satisfied with how the generative model is derived and presented. In my opinion, the problem could be expressed as a stochastic optimal control with an unknown cost function. However, the paper takes a different approach, starting from Bayesian optimization, which typically does not incorporate control or dynamics. This choice leads to confusion in notation and extends the Bayesian optimization framework. For instance, the notation between sections 2.1 and 2.2 changes due to the introduction of the action variable a. Additionally, in Section 2.1, the authors discuss a set of arms, drawing from the bandit literature.
What also confuses me is that the actions in the model are never formally introduced. Especially in Section 2.3, a formal definition of actions would have been helpful.

Background Section:

In the background section, it is not very obvious what the paper’s contribution is versus what constitutes background information. The citations are not very transparent.

Related Work (Section 2.4):

I believe one of the most important related concepts, Bayesian Reinforcement Learning or Dual Control, is not discussed at all.

Acquisition Function Analysis:

Additionally, I would have appreciated more analysis on why the specific acquisition function was chosen over others. How would the algorithm perform with a different acquisition function?

**Questions:**

- Is the utility function presented in the paper a contribution? Could the authors elaborate a bit on this?
- I am a bit confused about the notation in line 132. Could the authors please define $\tau_i$?
 - In line 136 and 199 is $S$ a switch in notation?
-	Could the authors elaborate a bit on line 209. What exactly is sampled using Thompson Sampling?
-	In line 259, why is this specific contstraint of 0.5 chosen?
-	I am a bit confused about line 263. I do not think MPC is nescessarly convex. There is a lot of literature on non-linear MPC. Could the authors elaborate on this?

**Limitations:**

-

---

> ### Author Rebuttal · Authors · 2024-08-06
>
> **W: Generally, I believe the paper’s presentation could be enhanced...**
>
> Indeed, the reviewer is right we could start by presenting the problem as a control problem with an unknown objective, and then derive a surrogate for the unknown objective that is refined over episodes. Our main audience is practitioners of Bayesian optimization; hence, we chose the route in the paper. We believe this comment reflects preferences in exposition, not on the correctness or soundness of the work.
>
> The actions are *formally* defined in Sec. 2.2. We could have introduced them already in Sec. 2.1, but it would clutter the notation to no benefit in exposition. We will make sure to improve clarity in Sec. 2.1.
>
> **W: In the background section, it is not very obvious what the paper’s contribution is versus what constitutes background information. The citations are not very transparent.**
>
> Please see the general rebuttal.
>
> **W: I believe one of the most important related concepts, Bayesian Reinforcement Learning or Dual Control, is not discussed at all.**
>
> Indeed, these are related but also crucially different. For example, Bayesian RL uses a reward function that is linear in the state-action visitation, while our utility is non-linear but convex. Dual control goes more in the direction of an unknown transition system, which we assume to be known here but indeed constitutes a very reasonable further work. We can add a small discussion of both concepts.
>
> **W: Additionally, I would have appreciated more analysis on why the specific acquisition function was chosen over others. How would the algorithm perform with a different acquisition function?**
>
> The objective was motivated by a goal to minimize the probability of a mistake. It was then refined to be computationally tractable with only one inequality in the derivation that makes it not tight (see general rebuttal). Our framework with the acquisition function as we derive it has the advantage that the induced state-action distribution of a trajectory allows for adaptive resampling and is theoretically grounded.
>
> If we had a different goal to maximizer identification, a different acquisition function would perhaps be more appropriate and could be used within our framework provided that it depends only on the state-action visitations akin to described in Mutny et. al. (2023) in Lemma 1. We leave this to future work.
>
> **Q: Is the utility function presented in the paper a contribution? Could the authors elaborate a bit on this?**
>
> Please see the general rebuttal.
>
> **Q: I am a bit confused about the notation in line 132. Could the authors please define $\tau_i$?**
>
> $\tau_i$ is the deployed trajectory at the $i$th episode. We will clarify this in the paper.
>
> **Q: In line 136 and 199 is $S$ a switch in notation?**
>
> This was a typo, thanks for pointing it out.
>
> **Q: Could the authors elaborate a bit on line 209. What exactly is sampled using Thompson Sampling?**
>
> We use Thompson Sampling to sample the elements of the candidate set of maximizers. In other words, it is used as a finite approximation of the uncertainty quantification provided by the continuous set $\mathcal{Z}_t$. That is, we create $K$ samples of the GP and use the optimum of each sample to create a set of $K$ potential maximizers.
>
> **Q: In line 259, why is this specific constraint of 0.5 chosen?**
>
> This is an arbitrary choice related to the code implementation, as we work in the search space $[-0.5, 0.5]$ (and normalize functions and data to those bounds). In reality, any bounds could be chosen, in which case the constraint becomes $x \in [a, b]$. Thanks for pointing this out; we will clarify this in the paper.
>
> **Q: I am a bit confused about line 263. I do not think MPC is necessarily convex. There is a lot of literature on non-linear MPC. Could the authors elaborate on this?**
>
> One thing is non-linear, another is non-convex. In fact, there is a lot of literature on both. Most MPC is studied with quadratic costs (non-linear), and with linear dynamics, then it is convex in the action variables. This is what is taught in graduate MPC courses. Of course, there is also non-convex non-linear MPC (common with mixed integer, for example), but this is more exotic and needs to be heuristically solved. We fall into this category as our cost is the *utility* which is non-convex in the action variables, but it can be heuristically solved.

---

> > ### Comment · Reviewer_2XmZ · 2024-08-11
> >
> > I have read the rebutal and thank the authors for their answers. I will keep my score and still vote for acceptance of the paper.
> >
> > I want to mention one minor clarification that non-convex MPC is far from being exotic. If you design a controller for a non-linear dynamical system, lets say for a toy example like a cart-pole, the MPC problem is solved using a direct method. When the dynamics are not linearized, this leads to a non-linear dynamics constraints, which is non-convex.

---

### Official Review · Reviewer_mJAD · 2024-07-11

**Soundness:** 2
**Presentation:** 2
**Contribution:** 2
**Rating:** 3
**Confidence:** 2

**Summary:**

The paper introduces a new Bayesian optimization problem that finds the optimal policy to optimize a black-box function subject to transition constraints on the query. The method works for both discrete and continuous Markov chains. The paper empirically demonstrates several practical applications in physical systems, electron laser calibration, and chemical reactor optimization.

**Strengths:**

The problem formulation is new, and the paper effectively motivates it using several practical applications. The solution is based on principled methods, with several experiments supporting the practicality of the problem.

**Weaknesses:**

The paper may be improved by clarifying several questions below.

**Questions:**

1. Section 2 is the background but there are no citations in section 2.1. Is this a new solution proposed by the paper, or is it based on existing work?

2. Line 575: Please explain why the proprotionality holds. Monotonicity does not imply proportionality (same ratio).

3. Line 587: Please explain the two inequalities.

4. Would the authors please clarify equation (7)? What does it mean by the expectation of a state $X_t$? What is $\pi$ in $d_{\pi}$? Z_t and GP posterior may change after every time step, so I have a hard time understand why U remains the same and the expectation is only taken over X_t in equation (7). Furthermore, as the utility depends on the time step (the GP posterior and Z_t), two visitations to the same state at different time steps have two different utilities. Then, how do we combine all visitations into $d_\pi$ (lines 143-144).

5. The pseudocode in Algorithm 1 is unclear. What is d_pi (there is no initialization of it)? Why does the update of $\hat{d}\_{t,h+1}(x)$ match equation (11)? Why are GP_{t+1} and Z_{t+1} only updated with X_{t,H} and Y_{t,H} (the observations from episode t, not all episodes from 1 to t)? What are X_{t,H} and Y_{t,H} (the time step is only until H-1).

6. The policy depends on the time step, so pi_{t,0} is only deployed at the first time step. How is the state visitation frequency of such a policy estimated well if it is only deployed at the 1st time step of each episode?

7. Would the authors please clarify equation (12) using the correct subscript of d (d_{t,h})?

8. Some minor remarks: In equation (11), should j be from 1 to t-1? Line 122: y(xh) should be y(xh,ah).

**Limitations:**

There are no potential negative societal impacts.

---

> ### Author Rebuttal · Authors · 2024-08-06
>
> **Q: Section 2 is the background but there are no citations in section 2.1. Is this a new solution proposed by the paper, or is it based on existing work?**
>
> Please see the general rebuttal.
>
> **Q: Line 575: Please explain why the proportionality holds. Monotonicity does not imply proportionality (same ratio).**
>
> This is a typo. With the first $A \propto B$ we refer to there existing a finite constant such that $A(x) = C B(x)$. Our goal is to find $\arg\min_x A(x)$, so we can equivalently minimize $\min \log B(x)$ to identify the minimizer of $A(x)$ as well, and this is what the second $\propto$ refers to, but it should be replaced with $\stackrel{!}{=}$ and explained in the text for more clarity. We apologize for the confusion. In addition, later in line 578, there is a typo; instead of $\propto$, it should read $\approx$, see the preceding line for an explanation.
>
> **Q: Line 587: Please explain the two inequalities.**
>
> The first inequality is missing a square (a typo, see 574 for the correct definition). The step is incorrectly explained and in fact holds only for large $T \gg 0$, or when $\mathbf{V}_T \gg \mathbf{I}$, where then $(\mathbf{V}_T + \mathbf{I})^{-1}\mathbf{V} \approx \mathbf{I}$.
>
> The second inequality follows by $\mathbf{V}_T = \mathbf{X}^\top \mathbf{X}$, push-forward identity, $\mathbf{V}_T \preceq \mathbf{V}_T + \mathbf{I}$ and monotonicity of the matrix inverse under Löwener ordering. The two are then substituted back to the original objective. Notice that the objective is as such only when $T$ is large (asymptotic regime).
>
> **Q: Would the authors please clarify equation (7)? What does it mean by the expectation of a state $X_t$? What is $\pi$ in $d_{\pi}$? $Z_t$ and GP posterior may change after every time step, so I have a hard time understanding why $U$ remains the same and the expectation is only taken over $X_t$ in equation (7). Furthermore, as the utility depends on the time step (the GP posterior and $Z_t$), two visitations to the same state at different time steps have two different utilities. Then, how do we combine all visitations into $d_\pi$ (lines 143-144).**
>
> In the discussion around Eq. (7), $X_t$ refers to the set of trajectories (see L132). We analyze and derive the algorithm in the *episodic* setting (trajectory feedback), where we execute a trajectory in full before updating the posterior (GP and $Z_t$). A trajectory is a random set, and therefore we must define a utility with respect to the expected trajectory when using a policy $\pi$. Each policy $\pi$ induces a state-action distribution, which tells us how likely we are to visit a state in a realization of a trajectory, which we denote $d_\pi$. To our benefit, as we show in the derivations, the objective depends only on $d_\pi$.
>
> For trajectory feedback, $Z_t$ does not change at every time-step, only after the whole trajectory is executed. When planning, we keep $Z_t$ fixed. Later we introduce *instant feedback*, where GP and policy are recalculated at each time step. However, we do not provide a formal derivation of optimality for this setup. The utility is still defined properly where we optimize the remaining part of the trajectory and by extension $\pi$ or $d_\pi$ or the time-steps from $h$ to $H$. This draws a parallel to the MPC literature of using receding horizon re-planning.
>
> **Q: The pseudocode in Algorithm 1 is unclear. What is $d_\pi$ (there is no initialization of it)? Why does the update of $\hat{d}\_{t,h+1}(x)$ match equation (11)? Why are $GP_{t+1}$ and $Z_{t+1}$ only updated with $X_{t,H}$ and $Y_{t,H}$ (the observations from episode $t$, not all episodes from 1 to $t$)? What are $X_{t,H}$ and $Y_{t,H}$ (the time step is only until $H-1$).**
>
> Note $d_\pi$ does not require initialization as it is the variable of the utility; it denotes the state-action distribution of a policy. In the second line of the algorithm you can see we take an argmax over this variable.
>
> Throughout the algorithm, we want to keep track of the visited states. We do this via the empirical state-action distribution, defined by the states we have visited so far. After visiting new state(s), we want to add them to the empirical state-action distribution which we can do by following the normalization procedure defined in eq. (11). For clarity, the empirical state-action distribution $\hat{d}\_{t,h}$ denotes _past_ states, while the variable $d_\pi$ is representing _future_ states we must choose.
>
> The GP and $Z$ are updated using observations from all previous trajectories. In the paper, we use lower-case variables to denote individual observations, and upper-case variables to denote all trajectories up to the time-step specified by the index.
>
> **Q: The policy depends on the time step, so $\pi_{t,0}$ is only deployed at the first time step. How is the state visitation frequency of such a policy estimated well if it is only deployed at the 1st time step of each episode?**
>
> The state-action frequency of a given policy can be calculated exactly; there is no need to deploy the policy itself. We assume the transition dynamics are known.
>
> **Q: Would the authors please clarify equation (12) using the correct subscript of $d (d_{t,h})$?**
>
> Notice that $\hat{d}_{t,h}$ refers to the empirical visitation. In other words, the frequency of visited states in trajectory $t$ at time horizon $h$. This visitation changes as we execute the policies for a longer time and more trajectories—this is updated depending on the trajectory we have followed up to time-step $t, h$. The empirical visitation $\hat{d}$ and also the variable $d$ in optimization Eq. (12) is $d:\mathcal{S} \times \mathcal{A} \times H \rightarrow [0,1]$.
>
> **Q: Some minor remarks: In equation (11), should $j$ be from 1 to $t-1$? Line 122: $y(xh)$ should be $y(xh,ah)$.**
>
> Yes, thanks for pointing out the mistakes.

---

> > ### Comment · Reviewer_mJAD · 2024-08-11
> >
> > Thank you for your response. However, I remain unclear about some of the notations and derivations presented in the paper.
> >
> > After reviewing the suggested lines 574-578, it appears that there is some confusion between proportionality and monotonicity within the proof. It occurs in several places, making it challenging to verify the correctness of the arguments, for instance, the expression following line 575, the subsequent omission of the log term, and the expectation taken over $f$ after line 578. Even if we replace "proportional to" with "approximate to," the quality of this approximation remains ambiguous. Moreover, the proof of Equation (14) is intended to establish a "tight upper bound" (as noted in line 94), rather than an approximation.
> >
> > The author posits that: A trajectory is a random set, and therefore we must define a utility with respect to the expected trajectory when using a policy $\pi$." However, this raises the question: why not take the expectation of the utility over the trajectories, similar to reinforcement learning approaches that maximize expected total reward over trajectories rather than the reward of the expected trajectory? Additionally, in the context of a discrete state space, how should we conceptualize the "expected trajectory"?
> >
> > Regarding Equation (7), $\pi$ is still unclear to me, does it refer to $\pi_1$​ or $\pi_2$ or something else​?

---

> > > ### Author Response · Authors · 2024-08-12
> > >
> > > Thanks to the reviewer for reading our responses carefully, and bringing up the following points of discussion. We will split our answer in two parts, first carefully providing details of the derivation, and secondly clarifications on the notation and concepts of eq. (7).
> > >
> > > **On the derivation of the objective:**
> > >
> > > Based on the discussion, L575 now reads:
> > > $$ P(\mu_T(z) - \mu_T(x_f^\star) \geq 0 | f) \propto \exp\left(-\frac{a_{z}^2}{b_{z}^2}\right)\frac{1}{b_{z}}\implies \log  P(\mu_T(z) - \mu_T(x_f^\star) \geq 0 | f) \propto \frac{b_z^2}{a_z^2} + \log(b_{z}) $$
> > > and L578:
> > > $$ E_{f\sim GP}[\log P(\mu_T(z) - \mu_T(x_f^\star) \geq 0 | f)] \leq \frac{1}{C} E_{f\sim GP}\left[\frac{b_z^2}{a_z^2}\right] $$
> > > where $C$ is a constant of proportionality, and the upper bound holds asymptotically as $T \rightarrow \infty$ or in fact for a sufficiently large $T$, (where $\log b_z \leq 0$ as long as $b_z \leq 1$, which has to eventually happen at finite $T$).
> > >
> > > Then the following lines (587) will read:
> > > \begin{eqnarray*}
> > > a_z^2 & = & (\theta^\top(V_T+I_{H})^{-1} V_T (\Phi(z) - \Phi(x^\star_f)))^2 \\ & \stackrel{T\gg 0}\approx & (\theta^\top (\Phi(z) - \Phi(x^\star_f)))^2 = (f(z) -  f(x^\star_f))^2\\
> > > \end{eqnarray*}
> > > which holds in the asymptotic regime as $(V_T + I_H)^{-1}V_T \approx I$. In fact, we can give order dependence for the specific instance where $V_T$ is made up from unit vectors. In this case, $a_z^2 = (f(z) -  f(x^\star_f))^2(1 - \mathcal{O}\left(\frac{1}{T}\right))$, so the gap rapidly goes down.
> > >
> > > The rest of the derivation remains the same. As mentioned in our general rebuttal, the objective we derived has been studied before. We chose the objective because of its asymptotic optimality properties and good empirical performance. We developed a new, Bayesian, motivation for this existing objective rather than provide a formalized theorem. The derivation's reliance on asymptotic arguments ($T\rightarrow\infty$) is in line with the frequentist studies (Fiez et al. 2019).
> > >
> > > We agree that the approximation quality remains unclear for small $T$. Claiming a "tight upper bound" is perhaps overselling, by this we meant that there is an instance where the above holds with equality. This is in fact true for any instance in the large $T$ limit.  We will add this note to the camera-ready, and remove the words "tight upper bound". However note, we do not use this upper bound for the algorithm. We clearly state we make yet another upper bound which is perfectly formal, and then use this in the algorithm. This second gap we formally analyze in Appendix C.5.

---

> > > ### Author Response · Authors · 2024-08-12
> > >
> > > **On eq. (7)**
> > >
> > > Our final objective will be a function of all executed trajectories. At each episode $t$, we deploy a policy $\pi_t$. $\pi$ refers to the aggregation of all policies $\pi: \\{1, ..., T\\} \times X \times A \rightarrow [0, 1]$.
> > >
> > > **Q: Why not take the expectation of the utility over the trajectories ... how should we conceptualize the "expected trajectory"?**
> > >
> > > Taking the expectation outside as $\mathbb{E}_{\tau\sim \pi}[F(\tau)]$ is intractable in nearly all cases, see Mutti et. al. (2023) for discussion in the context of reinforcement learning. Hence, all works studying remotely similar problem in formal fashion (providing tractable certificates) focus on the case where expectation is inside.
> > >
> > > By considering utilities of the expectation, we make the problem tractable (not NP-hard; even not NP-hard to approximate as in Prajapat et. al. (2024)). This follows the classical approach in the field of experiment design (Fedorov, 1997), convex reinforcement learning (Zahavy et. al., 2019) and prior work of Mutny et al. (2023), where the expectation is inside.
> > >
> > > The expected trajectory is not well conceptualized in our manuscript. As we state in L139, the concept borrows on the "expected trajectory" from Mutny et. al. (2023) on which this work builds up. We do not utilize the formalization of this anywhere later in the paper, hence we felt no need to reproduce it.
> > >
> > > Let us summarize here what we mean by $U$ (non-caligraphic), and $U(\mathbb{E}[X])$ in this context. The conceptualization begins by introducing the space of probability distributions over trajectories $\mathcal{P}$, here, $\eta_\pi(\tau) \in \mathcal{P}$ is a distribution describing the probability of obtaining trajectory $\tau$ upon deployment of policy $\pi$. Each policy $\pi$ induces $\eta_\pi(\tau)$. Similarly any finite execution of policy $\pi$ (say $n$ times) leading to observed trajectories $\mathbf{X}$ as in  Eq. (4) induces an empirical distribution $\hat\eta_\pi(\tau)$; this is the distribution where all our *observed* trajectories are given equal mass. The function $U$ is a function over probability distributions over trajectories $U:\mathcal{P}\rightarrow \mathbb{R}$, and for **all** empirical measures they coincide with $U(\mathbf{X}) =U(\hat\eta_\pi)$ from Eq. (4). Expectation of the empirical measure leads to $\mathbf{E}[\hat\eta_\pi] = \eta_\pi$, and this is the way to think about it. In particular, for a single episode, we write:
> > > $$
> > > \mathcal{U}(d_\pi) \stackrel{!} = U(\mathbb{E}[\mathbf{X}]) := U\left(E_{\tau \sim \pi} \left[ \delta_{\tau}\right]\right) = U \left(\sum_{\tau} \eta_{\pi}(\tau) \delta_{\tau} \right)
> > > $$
> > >
> > > Note that the sum is over all possible trajectories, making it intractable. For our objective, however, we can show it can be equivalently written in the variables of space of state-action distributions (!), we only have to sum over all states and actions which makes solving the problem solvable ($(|\mathcal{X}||\mathcal{A}|)^H$ to $|\mathcal{X}||\mathcal{A}|H$ variables). For completeness, for more episodes:
> > > $$
> > > \mathcal{U}(d_\pi) = U\left(E_{\tau_1 \sim \pi_1, ..., \tau_t \sim \pi_t}\left[\frac{1}{t}\sum_{i=1}^t \delta_{\tau_i}\right]\right) = U \left(\frac{1}{t} \sum_{i=1}^t\sum_{\tau} \eta_{\pi_i}(\tau) \delta_{\tau} \right)
> > > $$
> > > We will refer the readers to Appendix for the clarification of these concepts, but the most natural way to eliminate the confusion is most likely to introduce expected empirical visitations $\mathbb{E}[\hat{d}(x,a)]$ instead of $\mathbb{E}[\mathbf{X}]$. We have used $\mathbb{E}[\mathbf{X}]$ to make the connection to prior works, but its most likely causing confusion.
> > >
> > > References:
> > >
> > > - Mutti e.t al. (2023), Convex Reinforcement Learning in Finite Trials, JMLR
> > >
> > > - Zahavy et. al. (2021), Reward is enough for convex MDPs, NeurIPS 2021
> > >
> > > - Fedorov (1997), Model-Oriented Design of Experiments, Springer
> > >
> > > - Prajapat et. al. (2024), Submodular Reinforcement Learning, ICLR 2024

---

> > > > ### Comment · Area_Chair_V1qm · 2024-08-12
> > > >
> > > > Reviewer mJAD, could you please respond do the additional comments made by the authors? Currently there is a rather large discrepancy between your score and the other reviewers, and your review didn't point to any major weaknesses other than clarity (given your questions). I would like to understand whether the authors' additional responses helped clarify your confusion. Thanks.

---

> > > > > ### Comment · Reviewer_mJAD · 2024-08-13
> > > > >
> > > > > Thank you for your clarification. I still have a few questions, and I would appreciate your assistance in understanding the following points.
> > > > >
> > > > > **On the derivation of the objective**
> > > > > Let us go through your derivation step by step.
> > > > >
> > > > > $P(\mu_T(z) - \mu_T(x^*_f) \ge 0|f) \propto \exp (- \frac{a_z^2}{b_z^2}) \frac{1}{b_z} \Rightarrow P(\mu_T(z) - \mu_T(x^*_f) \ge 0|f) = C_1 \exp (- \frac{a_z^2}{b_z^2}) \frac{1}{b_z}$
> > > > >
> > > > > for some constant $C_1 \neq 0$. By taking the logarithm of both sides,
> > > > >
> > > > > $\log P(\mu_T(z) - \mu_T(x^*_f) \ge 0|f) = \log C_1 - \frac{a_z^2}{b_z^2} + \log \frac{1}{b_z} = \log C_1 - \frac{a_z^2}{b_z^2} - \log b_z$
> > > > >
> > > > > Then, how does it imply
> > > > >
> > > > > $\log P(\mu_T(z) - \mu_T(x^*_f) \ge 0|f) \propto \frac{b_z^2}{a_z^2} + \log b_z$
> > > > >
> > > > > and
> > > > >
> > > > > $E_{f \sim GP} [...] \le \frac{1}{C} E_{f \sim GP} [\frac{b_z^2}{a_z^2}]$ (note that we still have a constant $\log C_1$)
> > > > >
> > > > >
> > > > > **On expected trajectory**
> > > > > I have just skimmed through the work of Mutti et al. (2023) mentioned in the author response, so please correct me if I'm wrong. From what I read, they first review the original RL formulation that maximizes the expected total reward (not the reward of the expected trajectories) (Equation (1) in Mutti et al. 2023), which can be rewritten in terms of the state distribution induced by the policy. They then discuss a generalization of the RL formulation (Equation (2) in Mutti et al. 2023) by directly introducing a utility function based on the state distribution induced by a policy. It is important to note that this utility function is defined for a state distribution (or its empirical distribution), rather than for a specific trajectory (the expected trajectory).
> > > > >
> > > > > From the author’s response, I now understand that what the author mean may be the empirical state distribution as described in Mutti et al. (2023). I believe the paper's notation should be revised to reflect this.
> > > > >
> > > > > However, this also raises a concern: As the author responded,
> > > > >
> > > > > $U(d_\pi) = U(\sum_{\tau} \eta_{\pi}(\tau) \delta_\tau)$
> > > > >
> > > > > In Equation (4), $U$ can only take trajectories (or a set of inputs) as it represents the posterior variance of $f(z) - f(z')$ given the observations. When considering $U(\sum_{\tau} \eta_{\pi}(\tau) \delta_\tau)$, it is unclear how to evaluate $U$ for $\sum_{\tau} \eta_{\pi}(\tau) \delta_\tau$​, which is a weighted average (empirical distribution) of the inputs. This raises the question of how to evaluate the posterior variance of $f$ given an empirical distribution of the inputs. Furthermore, Lemma D.1 also pertains to the posterior variance given a set of inputs $X$, rather than an empirical distribution of the inputs.

---

> > > > > > ### Author Response · Authors · 2024-08-13
> > > > > > **Response to reviewer 2 / 2**
> > > > > >
> > > > > > > From the author’s response, I now understand that what the author mean may be the empirical state distribution as described in Mutti et al. (2023). I believe the paper's notation should be revised to reflect this.
> > > > > >
> > > > > > We happily agree, this will improve clarity.
> > > > > >
> > > > > > > However, this also raises a concern...
> > > > > >
> > > > > > There seems to be confusion between empirical visitation and a visitation as a variable. For simplicity, we will only talk about state visitations, but the more complete treatment with state-action visitations is analogous.
> > > > > >
> > > > > > There is a $d_\pi$ a variable from the (state) visitation polytope, and then there is empirical visitation of the states after a policy has been executed $\hat{d}$. The notation we follow is that $\hat{d}$ are empirical ones. All objectives are functions of $d_\pi$ and we optimize over it, and it leads to convex programs in this variable. Sometimes the objective features $\hat{d}$ serving as an offset for the objective, not a variable.
> > > > > >
> > > > > > For experiment design we propose the optimization of Eq.(8) given by:
> > > > > > \begin{eqnarray}
> > > > > >       \mathcal{U}(d_\pi) = \max_{z, z' \in \mathcal{Z}} || \Phi(z) - \Phi(z')||^2_{V(d_\pi)^{-1}}
> > > > > > \end{eqnarray}
> > > > > > where $V(d_\pi) = \left( \sum_{x \in \mathcal{X}} \frac{d_\pi(x)\Phi(x) \Phi(x)^T}{\sigma^2(x)} + I \right)$. This is an objective with a state distribution $d_\pi$ as a variable. Now assume we sample a trajectory $X$. In the objective introduced in Eq.(4), we showed in Lemma D.1 that we can equivalently write it as:
> > > > > > \begin{equation}
> > > > > >     U(X) = \max_{z, z' \in \mathcal{Z}} \text{Var}[f(z) - f(z') | X] = \max_{z, z' \in \mathcal{Z}} || \Phi(z) - \Phi(z')||^2_{V(X)^{-1}}
> > > > > > \end{equation}
> > > > > >
> > > > > > where $V(X) = \left( \sum_{x \in \mathcal{X}} \frac{\hat{d}_X(x)\Phi(x) \Phi(x)^T}{\sigma^2} + I \right)$. In particular, note that we can write $V(X)$ (and hence $U(X)$) solely as a function of the state-distribution of the trajectory $\hat{d}_X$. Most importantly, the value is exactly the same as the objective in Eq.(8) where we evaluated $\mathcal{U}(\hat{d}_X)$, i.e. setting d_pi to the empirical distribution of trajectory $X$. As the reviewer points out, in this case, not every state distribution translates to a trajectory, but this is not an issue for our objective.
> > > > > >
> > > > > > Secondly, when we optimize over state-action polytope $d$, we are in essence doing a continuous relaxation of a discrete optimization over empirical visitations with a fixed $T$. Once we have an optimal state density, we are able to build a policy from it. And once we have a policy we can deploy it in the environment to obtain a new trajectory. If it helps, it is similar to solving an integer optimization problem by considering a continuous relaxation of a function. The "rounding" in the relaxation proceeds via sampling from the policy for our case. It is an analog of convex relaxations in the experiment design literatures. For more examples, see the book on convex optimization (Boyd and Vandenberghe , 2003) or for something more relevant to our work, Mutny et. al. (2023), where they study different methods of rounding.

---

> > > > > > > ### Comment · Reviewer_mJAD · 2024-08-14
> > > > > > >
> > > > > > > Thank you for the clarification. The corrected proof resolves my concern. However, the explanation regarding $\mathcal{U}(d_{\pi})$, where $d_{\pi}$ is from the visitation polytope (normalized), and the original definition of $U(X)$ in Eq. (4)  and Lemma D.1 ($X$ is considered as "count") is still unclear to me.
> > > > > > >
> > > > > > > As I mentioned in my previous comment, $U(X)$ represents the posterior variance of $f(z) - f(z') | X$. How do we then compute the posterior variance of $f(z) - f(z') | d_{\pi}$ (given a state visitation)? If I understand correctly, the expressions seem not equivalent. Specifically, $V(d_{\pi})$ involves a normalized $d_{\pi}$, while $V(X)$ involves an unnormalized $\hat{d}_X$. For them to be equivalent (or proportional for optimization purposes), we would need to multiply $d\_{\\pi}$ by some constant and check if $\\mathcal{U}(d\_{\\pi})$ is proportional to $U(d\_{\\pi} \\times \\text{constant})$ (such that $d\_{\\pi} \\times \\text{constant}$ is un-normalized), where $U(d\_{\\pi} \\times \\text{constant})$ is evaluated using Eq. (4).
> > > > > > >
> > > > > > > Could the author provide further explanation on this or derive $\\mathcal{U}(d\_\pi)$ from Eq.(4)?

---

> ### Author Response · Authors · 2024-08-13
> **Response to reviewer 1 / 2**
>
> Thank you the response and for carefully analyzing our derivation. We want to again mention that the objective used in our paper has been used in works of Fiesz et. al. (2019) published at NeurIPS. We hope there is no doubt that this is a meaningful objective. We only provide a Bayesian motivation for it.
>
> Upon a more meticulous look of our derivation, we found we had an unnecessarily complicated path that led to the sign blunder and wrong fraction order that the reviewer points out (which ended up cancelling each other out). Allow us instead to provide a more elegant and simple derivation (which will be added to the revised paper). It follows from the Gaussian tail bound that states for any $X \sim N(\mu, \sigma^2)$ and $t > 0$ then:
> $$ \mathbb{P}(X \geq \mu + t) \leq e^{-\frac{t^2}{2 \sigma^2}} $$
> We can now bound the probability of making an error, indeed $\mu_T(z) - \mu_T(z^*) \sim N(a_z, b_z^2)$:
> $$ \mathbb{P}(\mu_T(z) - \mu_T(z^*) \geq 0) = \mathbb{P}(\mu_T(z) - \mu_T(z^*) \geq a_z + (-a_z) ) \leq e^{-\frac{a^2}{2 b^2}}$$
> with the caveat that the bound only holds asymptotically as $T \rightarrow \infty$, as in this regime $a_z \approx f(z) - f(x_f^*)$, due to asymptotic argument we made previously. Notice that by definition of $x_f^*$, $f(z) - f(x_f^*)\leq 0$, hence substituting for $t = -a_z$,
> \begin{equation}
>     E_{f\sim GP}[\log P(\mu_T(z) - \mu_T(x_f^\star) \geq 0 | f)] \leq - \frac{1}{2} E_{f\sim GP}\left[\frac{a_z^2}{b_z^2}\right]
> \end{equation}
> by taking first the log, and second the expectation on both sides. From here the arguments follow in a similar fashion to the original paper, with the caveat that originally we had an error sign and the numerator and denominator the wrong way around and so the errors cancelled out. For full clarity here is the argument:
>
> Our aim is now to minimize making an error, so minimizing $-E_{f \sim GP}\left[\frac{a_z^2}{b_z^2}\right]$. It then follows that in the large $T$ limit (from the previously discussed bounds on $b_z$ and $a_z \approx f(z) - f(x_f^\star)$):
> \begin{equation}
>     - E_{f\sim GP}\left[\frac{a_z^2}{b_z^2}\right] \leq - E_{f\sim GP} \left[ \frac{(f(z) - f(x_f^\star))^2}{k_{\mathbf{X}}(z, x_f^\star)} \right]
> \end{equation}
> combining the two equations we obtain:
> \begin{equation}
>     E_{f\sim GP}[\log P(\mu_T(z) - \mu_T(x_f^\star) \geq 0 | f)] \leq - \frac{1}{2} E_{f\sim GP} \left[ \frac{(f(z) - f(x_f^\star))^2}{k_{\mathbf{X}}(z, x_f^\star)} \right]
> \end{equation}
> note the correct interpretation of the bound: the probability of an error is small if (i) the uncertainty is small, or (ii) the difference in values is large. Then we simply follow the arguments in Section C.3 to arrive to the objective:
> \begin{equation}
>     - E_{f\sim GP} \left[ \frac{(f(z) - f(x_f^\star))^2}{k_{\mathbf{X}}(z, x_f^\star)} \right] \leq - \frac{E_{f\sim GP}[\text{gap(f)}]}{\text{Var}[f(z) - f(z') | \mathbf{X}]}
> \end{equation}
> And finally, we note that due to the negative sign, and the denominator being constant, minimizing the RHS is equivalent to minimizing the objective introduced in Eq.(4), i.e. $\arg\min_{x}-g(x)$  is equivalent to $\arg\min_{x} \frac{1}{g(x)}$ when $g(x) > 0$.

---

> ### Author Response · Authors · 2024-08-14
> **Response to reviewer**
>
> We would like to thank the reviewer for their prompt response.
>
> > The corrected proof resolves my concern
>
> We are happy to hear that. Please consider raising your score, especially if you find the following clarifications useful.
>
> > However, the explanation regarding...
>
> The reviewer's intuition is correct, let us explain in detail what this "constant" is and how we use it (in essence the constant is the total length of the trajectories):
>
> Firstly, we would like to clarify that empirical state visitations are also normalized. This can be seen, for example, in Eq.(11) where we introduced them.
>
> Let us also note that the posterior variance of a Gaussian process does not depend depend on observed values, but only the points $x$, hence we can forward plan reduction in uncertainty. This is why Eq. (4) is equal to Eq. (6). Now, we show that the relaxation of the objective to state distributions holds in the simplest setting, for notation we will use $X$ to refer to the trajectory and $S$ to refer to the state space (in the paper we use $\mathcal{X}$):
>
> For simplicity assume we have a single trajectory of length $H$ and no observation noise:
> \begin{equation}
>     V(X) = \sum_{x \in {X}} \Phi(X) \Phi(X)^T = H \sum_{x \in S} \hat{d_X} (x) \Phi(X) \Phi(X)^T:= H V(\hat{d_X})
> \end{equation}
> Now we note that:
> \begin{equation}
>    ||\Phi(z) - \Phi(z')|| \_{V(X)^{-1}} = (\Phi(z) - \Phi(z'))^T V(X)^{-1} (\Phi(z) - \Phi(z'))
> \end{equation}
> \begin{equation}
> = \Phi(z)^T V(X)^{-1} \Phi(z) + \Phi(z')^T V(X)^{-1} \Phi(z') - 2 \Phi(z)^T V(X)^{-1} \Phi(z')
> \end{equation}
> \begin{equation}
> = \text{Var}[f(z) | X] + \text{Var}[f(z') | X] - 2 \text{Cov}[f(z), f(z') | X] \\
> \end{equation}
> \begin{equation}
> = \text{Var}[f(z) - f(z') | X]
> \end{equation}
> Note that since $V(X) \propto V(\hat{d}\_X)$, then:
> \begin{equation*}
>     \text{Var}[f(z) - f(z') | X] \propto ||\Phi(z) - \Phi(z')||\_{V(\hat{d}\_X)^{-1}}
> \end{equation*}
> so the optimization of the objectives is equivalent, note that the proportionality constant is exactly the normalization constant. In Lemma D.1 we formalize this more carefully in the setting with many trajectories and observation noise. We have to change the regularization of the objective appropriately (e.g. we must scale the contribution of the prior by $1 / H$, when planning for one episode, to counteract the normalization of the state-distributions), i.e., for the objectives to be equivalent we must use:
> \begin{equation*}
>     V(d_\pi) = \left(\sum_{x \in S} d_\pi(x) \frac{\Phi(x)^T \Phi(x)}{\sigma^2} + \frac{1}{H} I \right) = \frac{1}{\sigma^2}\left(\sum_{x \in S} d_\pi(x) \Phi(x)^T \Phi(x)+ \frac{\sigma ^2}{H} I \right).
> \end{equation*}
> For future reference, there are still two minor typos in the paper that we found, Eq. (8) it should read:
> \begin{equation*}
>     V(d_\pi) = \left(\sum_{x \in S} d_\pi(x,a) \frac{\Phi(x)^T \Phi(x)}{\sigma^2(x,a)} + \frac{1}{TH} I \right).
> \end{equation*}
> where we plan for $T$ episodes (trajectories) to make it consistent with the correct result in Lemma D.1. And in the proof of Lemma D.1, in the last equation of L705 there should not be a $TH$ before the sum. We wanted to manipulate $TH$ out but forgot it inside the inverse -- an honest typo. We will fix the typos and add a clarification that we need the correct regularization for the objectives to be equivalent.
>
> To be fully clear, our implementation of the objective in the code does include the correct regularization procedure (see file mdpexplore/functionals/bandit\_functionals.py L242 where we multiply the identity by $1 - \alpha$).

---

> > ### Comment · Reviewer_mJAD · 2024-08-14
> >
> > Thank you for the clarification. As I expected, $\mathcal{U}(d_\pi)$ is not proportional to the expression derived from Eq. (4) due to the noise term in $V$. This issue affects the main objective of the paper, and without knowing whether $1 - \alpha$ term is the same as $1 / TH$, I am unable to verify its correctness. Considering other errors in the proof and notations, my assessment of the paper remains unchanged.

---

> > > ### Author Response · Authors · 2024-08-14
> > >
> > > We disagree with the reviewer’s assessment, we do show that $\mathcal{U}(d_\pi)$ (as defined in the Appendix, and after fixing the typo in Eq. (8)) is proportional to Eq. (4). We stand by correctness of our work and the theoretical and empirical results we presented.
> > >
> > > We would like to thank the reviewer for their time.

---

### Official Review · Reviewer_J8j4 · 2024-07-12

**Soundness:** 4
**Presentation:** 4
**Contribution:** 3
**Rating:** 7
**Confidence:** 3

**Summary:**

&nbsp;

The authors address the problem of transition-constrained Bayesian optimization, modeling the transition constraints as a Markov decision process. They take a novel approach in deriving their utility function for this problem setting based on maximum identification and bounding the probability of choosing an incorrect maximizer. While the method is well-described and has the potential to yield theoretical insight into the structure of transition-constrained Bayesian optimization, the empirical performance does not seem hugely compelling compared to existing methods such as SnAKe and LSR. As such, my recommendation is borderline with potential to increase my score if the authors can address the main criticism of the advantages of the method compared to competitor approaches.

&nbsp;

**Strengths:**

&nbsp;

The authors derive a novel transition-constrained Bayesian optimization algorithm, leveraging the literature on maximum identification via hypothesis testing to derive their utility function.

&nbsp;

**Weaknesses:**

&nbsp;

__MAJOR POINTS__

&nbsp;

1. The main weakness of the work would appear to be the lack of compelling evidence for why the authors' method should be chosen over competitor methods such as LSR or SnAKe on practical problems. I think the paper would benefit substantially if the authors could highlight problem settings in which they expect their algorithm to perform better in relation to at least one practical performance metric.

2. The code would benefit from a README describing how to reproduce the results of the paper.

&nbsp;

__MINOR POINTS__

&nbsp;

1. It would be great if the references appeared in numbered order.

2. There is some missing capitalization in the references e.g. "bayesian" in place of "Bayesian".

3. Line 90, typo, "we seek to identify".

4. Line 566, typo, "due to selecting".

5. Line 566/567, missing full stop at the end of the equation.

6. On line 567 epsilon is homoscedastic but in the main paper the authors state that epsilon can be heteroscedastic?

7. Line 584, typo, "hypotheses".

8. Line 680, typo, extraneous "be".

9. Line 680, in the notation should there be some kind of indication that X corresponds to T trajectories e.g. an indication of the dimensions of the space X corresponds to? The same on line 690.

10. In Equation 4, the conditioning bar is difficult to read, could the authors introduce some more space to the left and right of the bar?

11. In the Subsection on "Utility with Embeddings", the authors may wish to specify "Kernel Embeddings" so as to disambiguate from embeddings over the input space X.

12. In the proof of statement 25, it might be worth giving the acronym for SMW, namely Sherman-Morrison-Woodbury, somewhere in the text.

13. On line 136, have the authors defined S? This is the state space presumably.

14. In the related work on look-ahead BayesOpt, it may also be worth mentioning [1].

15. Line 179, typo, "a mix of the previous two".

16. In Section 2.2, the variable d could be explained in more detail.

17. When referencing Thompson sampling, it would be worth citing the original paper [2].

18. In Section E.2, the expression for the RBF kernel implicitly assumes x is 1-dimensional. The authors could replace the squared distance with the squared 2-norm to generalize the expression.

19. In Algorithm 1, would it be more descriptive to refer to Z_0 as the initial set of maximizer candidates rather than the initial set of maximizers?

20. It would be worth citing [3] (as per the discussion the historical section by Garnett [4]) for the Expected Improvement acquisition function given that it is used.

21. It would be great if the authors could provide the number of random trials over which the error bars were computed in Figure 4.

22. Line 622, "scales them properly". What is "them"?

23. Line 670, typo, "using".

&nbsp;

__REFERENCES__

&nbsp;

[1] Roman Marchant, Fabio Ramos, and Scott Sanner. Sequential Bayesian optimisation for spatial-temporal monitoring. In Proceedings of the International Conference on Uncertainty in Artificial Intelligence, 2014.

[2] W. R. Thompson. On the likelihood that one unknown probability exceeds another in view of the evidence of two samples. Biometrika, 25(3-4):285–294, 1933.

[3] VR Saltenis (1971). One Method of Multiextremum Optimization. Avtomatika i Vychislitel’naya Tekhnika (Automatic Control and Computer Sciences) 5(3):33–38.

[4] Garnett, R., 2023. Bayesian optimization. Cambridge University Press.

&nbsp;

**Questions:**

&nbsp;

1. In Equation 3, do the authors have any intuition as to why k(z, x_f*) / (f(z) - f(x_f*))^2 is related to the probability of returning an wrong maximizer? Beyond the derivation presented in Section C.1 of the appendix.

2. In Equation 4, the authors state their objective as a maximization problem. However, in the text, the authors refer to minimizing the uncertainty among all pairs in \mathcal{Z}. Could the authors explain this?

3. In Section D.1 of the appendix, the authors give the formulation of the objective for general kernel matrices. The authors state that the objective may be computed by inverting a |\Chi| x |\Chi| matrix. How restrictive is this for the scale of problems that can be addressed with this formulation of the objective?

4. In Section 5, what was the motivation for the decision to fix the GP hyperparameters?

5. For Figures 2 and 3 why are there no error bars for the average regret curves?

&nbsp;

**Limitations:**

&nbsp;

1. The main limitation as described above would appear to be the lack of compelling evidence for problem settings under which the authors' algorithm outperforms other transition-constrained Bayesian optimization methods such as LSR and SnAKe.


&nbsp;

---

> ### Author Rebuttal · Authors · 2024-08-06
>
> **W: The main weakness of the work would appear to be the lack of compelling evidence for why the authors' method should be chosen over competitor methods such as LSR or SnAKe on practical problems...**
>
> Please see the general rebuttal.
>
> **W: The code would benefit from a README describing how to reproduce the results of the paper.**
>
> We agree and we will add this after peer review. We did have a README but hastily deleted it to preserve anonymity!
>
> **W: On line 567 epsilon is homoscedastic but in the main paper the authors state that epsilon can be heteroscedastic?**
>
> For simplicity, the derivation is done for the homoskedastic setting, however the same derivation holds in the heteroskedastic setting by simply changing $\mathbf{V} = \sum_{i=1}^T \frac{1}{\sigma_i^2}\Phi(x_i)\Phi(x_i)^\top$ instead of for heteroskedastic least squares $\mathbf{V} = \sum_{i=1}^T \frac{1}{\sigma^2}\Phi(x_i)\Phi(x_i)^\top =: \mathbf{X} \mathbf{\Sigma}\mathbf{X}$. We can add this comment to the discussion.
>
> **W: Other minor points**
>
> Thanks for pointing out minor issues which we will address.
>
> **Q: In Equation 3, do the authors have any intuition...**
>
> It allows an interpretation as noise to signal ratio. In the numerator, we have uncertainty (covariance) between $f(z)$ and $f(x_f^*)$, while in the denominator we have the level of signal that is the gap between $f(z)$ and $f(x_f^*)$. The smaller the gap, the harder it is to know which of the two is maximal, increasing the probability of error. On the other hand, if the two states $x_f^*$ and $z$ are similar to each other, we do not need to put additional effort as they are very correlated, hence balancing these two is the right metric. We will add this intuition to the paper to improve clarity.
>
> **Q: In Equation 4, the authors state their objective as a maximization problem. However, in the text, the authors refer to minimizing the uncertainty among all pairs in $\mathcal{Z}$. Could the authors explain this?**
>
> The ultimate goal is to maximize an unknown function $f$. For exposition, as is common in BayesOpt, we talked about utility or acquisition function maximization, but then deriving the utility we find we need to minimize the uncertainty among all pairs. This lends itself to explain as minimization better. We could equally talk about maximization of utility $=$ 1/loss.
>
> **Q: In Section D.1 of the appendix, the authors give the formulation of the objective for general kernel matrices. The authors state that the objective may be computed by inverting a $|\mathcal{X}| \times |\mathcal{X}|$ matrix. How restrictive is this for the scale of problems that can be addressed with this formulation of the objective?**
>
> Indeed, this is a limitation seen also with some classical BO approaches such as Thompson sampling which need inversion of $|\mathcal{X}| \times |\mathcal{X}|$ matrix -- same complexity as our method.
>
> Common remedies are some finite dimensional approximations of the kernels, leading to $m \times m$ inversions or other reformulations improving the efficiency of matrix inversion that we cite in our work. Akin to Thompson sampling, even combinations of different techniques such as feature approximation and different sampling, namely Matheron rule for sampling, could be a very practical remedy (Wilson et al. 2020). This issue is a fundamental difficulty associated with reasoning (planning) in non-parametric function spaces, not necessarily something specific to our method. Also notice that in continuous domains the reparametrization of $d$ eliminates this inversion but leads to potentially non-convex program in the reparametrization.
>
> **Q: In Section 5, what was the motivation for the decision to fix the GP hyperparameters?**
>
> Please see the general rebuttal.
>
> **Q: For Figures 2 and 3 why are there no error bars for the average regret curves?**
>
> We did not include them to avoid cluttering the graphics. The percentage of runs we identify the maximizer allows us to see the robustness of each method. Nonetheless, we are happy to include complete regret plots in the appendix.
>
> **L: The main limitation as described above would appear to be the lack of compelling evidence for problem settings under which the authors' algorithm outperforms other transition-constrained Bayesian optimization methods such as LSR and SnAKe.**
>
> Please see the general rebuttal.

---

> > ### Comment · Reviewer_J8j4 · 2024-08-12
> > **Many Thanks to the Authors for their Rebuttal**
> >
> > &nbsp;
> >
> > Many thanks to the authors for their clarifications. For figures 2 and 3 it would have been nice to see the errorbars in the 1-page rebuttal document that could have been attached to the rebuttal comment.
> >
> > The major point of clarification stems from the problem settings under which the authors' method can be applied relative to LSR and SnAKe.  Given that this point has been fully addressed, I am raising my score.
> >
> > &nbsp;

---

> ### Comment · Area_Chair_V1qm · 2024-08-11
>
> Reviewer J8j4, could you please review the authors' response and see whether it addresses your open questions? Please acknowledge having done so in a comment together with a discussion of whether / how it has affected your assessment of the submission. Thanks.

---

### Official Review · Reviewer_MNRf · 2024-07-13

**Soundness:** 3
**Presentation:** 2
**Contribution:** 3
**Rating:** 6
**Confidence:** 3

**Summary:**

The paper studies the best arm identification task in the transition-constrained setting where the MDP captures the transition. To solve the resulting planning problem defined as maximizing the acquisition propogated through the MDP, the paper studies using the RL solvers and heuritics in the discrete and continuous cases.

**Strengths:**

1. Valid scenario, well-motivated and well-illustrated algorithm.
2. Comprehensive empirical evidence substantiating the effectiveness of the proposed algorithm, along with verification of parameter choices.

**Weaknesses:**

1. Although employing an RL solver, as admitted by the authors, the objective is non-convex and differs from conventional optimization in RL. This demands heuristics, which are not sufficiently discussed.

2. In the experiments, the GP hyperparameters are fixed, which is questionable due to potential misspecification and manually introduced bias. A typical practice might include either a full Bayesian treatment of the GP hyperparameters or kernel learning. This is particularly relevant for the ODE kernel due to its higher complexity.

3. The results in Figure 4 appear to lack statistical significance. Additionally, there is no clear preference among the variants of MDP-BO, resonating with the challenge in conventional BO.

4. There is a series of works on applying BO to graphical structures. Although the MDP formulation is different, adding them to the discussion would enhance completeness as some design choices are conceptually related.

***References***

1. Kusakawa, Shunya, Shion Takeno, Yu Inatsu, Kentaro Kutsukake, Shogo Iwazaki, Takashi Nakano, Toru Ujihara, Masayuki Karasuyama, and Ichiro Takeuchi. "Bayesian optimization for cascade-type multistage processes." Neural Computation 34, no. 12 (2022): 2408-243.
2. Aglietti, Virginia, Xiaoyu Lu, Andrei Paleyes, and Javier González. "Causal Bayesian optimization." In International Conference on Artificial Intelligence and Statistics, pp. 3155-3164. PMLR, 2020.
3. Astudillo, Raul, and Peter Frazier. "Bayesian optimization of function networks." Advances in Neural Information Processing Systems 34 (2021): 14463-14475.

**Questions:**

1. Could the author comment on the practice of limiting the extension of MDP-BO to either TS or UCB?
2. One minor question: in figure 3(a), is the legend "MDP-BO" written as "MDP-B0"?
3. The objective defined in Section 2.1 and the general BO framework presented in Algorithm 1, beyond the MDP-based planning solver, are commonly seen in previous works. Li and Scarlett (2022) studied the batched version of this formulation, Zhang et al. (2023) studied the high-dimensional treatment, and Han et al. (2024) studied the formulation in game theory, among others. There seems to be potential to generalize the proposed MDP-BO framework into broader applications. It would be a bonus point to additionally discuss the corresponding potentials of MDP-BO.

4. Moreover, recent advancements by Salgia et al. (2024) suggest that random exploration could be sufficient on $Z_t$. I encourage the authors to explore this if it is of interest.

***References***

1. Li, Zihan, and Jonathan Scarlett. "Gaussian process bandit optimization with few batches." In International Conference on Artificial Intelligence and Statistics, pp. 92-107. PMLR, 2022.
2. Zhang, Fengxue, Jialin Song, James C. Bowden, Alexander Ladd, Yisong Yue, Thomas Desautels, and Yuxin Chen. "Learning regions of interest for Bayesian optimization with adaptive level-set estimation." In International Conference on Machine Learning, pp. 41579-41595. PMLR, 2023.
3. Han, Minbiao, Fengxue Zhang, and Yuxin Chen. "No-Regret Learning of Nash Equilibrium for Black-Box Games via Gaussian Processes." In The 40th Conference on Uncertainty in Artificial Intelligence.
4. Salgia, Sudeep, Sattar Vakili, and Qing Zhao. "Random Exploration in Bayesian Optimization: Order-Optimal Regret and Computational Efficiency." In Forty-first International Conference on Machine Learning.

**Limitations:**

Discussed above.

---

> ### Author Rebuttal · Authors · 2024-08-06
>
> **W: The objective is non-convex and differs from conventional optimization in RL. This demands heuristics, which are not sufficiently discussed.**
>
> This is true for the continuous case and common to most setups in control theory. However, special to our setting, for all discrete environments in the experiments we are able to solve the sub-problems provably with convex solvers, see Frank-Wolfe updates.
>
> **W: In the experiments, the GP hyperparameters are fixed...**
>
> Please see the general rebuttal.
>
> **W: The results in Figure 4 appear to lack statistical significance. Additionally, there is no clear preference among the variants of MDP-BO, resonating with the challenge in conventional BO.**
>
> Please see the general rebuttal.
>
> **W: There is a series of works on applying BO to graphical structures...**
>
> Causal BO uses graphs that relate different variables in the optimization process according to a Bayesian causal model. Our graphs relate transitions between states. This is different in spirit, but we will add to the discussion.
>
> **Q: Could the author comment on the practice of limiting the extension of MDP-BO to either TS or UCB?**
>
> We obtained good results with both, so we did not try to use any other batch algorithms for estimating the maximizer candidate set. However, there is flexibility in the algorithm to use different estimation methods and could be the focus of future work.
>
> **Q: One minor question: in figure 3(a), is the legend "MDP-BO" written as "MDP-B0"?**
>
> Correct, this is an error, thanks for catching it!
>
> **Q: The objective defined in Section 2.1 and the general BO framework presented in Algorithm 1, beyond the MDP-based planning solver, are commonly seen in previous works. Li and Scarlett (2022) studied the batched version of this formulation, Zhang et al. (2023) studied the high-dimensional treatment, and Han et al. (2024) studied the formulation in game theory, among others. There seems to be potential to generalize the proposed MDP-BO framework into broader applications. It would be a bonus point to additionally discuss the corresponding potentials of MDP-BO.**
>
> Thanks for all the references, they are indeed relevant. We will include a discussion of them in the paper.

---

> > ### Comment · Reviewer_MNRf · 2024-08-11
> >
> > I appreciate the authors' response. I maintain my original evaluation that the setting and algorithm are novel and valid. Though my fellow reviewers have raised clarity issues, I don't foresee significant soundness problems. Hence, I'll keep my score.

---

### Author Rebuttal · Authors · 2024-08-06

We would like to thank all reviewers for the time spent reading and evaluating our paper. We are happy to have the strengths of our work recognized, and we are also pleased to have received so much constructive feedback. Here we address points common to multiple reviewers.

**Comparison with SnAKe and LSR:** Both SnAKe and LSR are algorithms specializing in a very specific type of transition constraint, when we have a restriction on how far we can move between experiments. The goal of our work is to deal with transition constraints in a more general and formalized setting. In fact, all the real-world experiments we consider are examples that SnAKe and LSR cannot handle. When considering local search constraints, it is not surprising that SnAKe and LSR perform strongly since they are specialist heuristics in this setting; however, we find it very encouraging that MDP-BO is able to match these SOTA algorithms.

**Section 2.1:** We included this section as background because similar objectives have been derived and used in the literature. Indeed, Fiez et al. [1] used the same objective for transductive linear bandits and there are other examples as reviewer MNRf pointed out. However, the motivation and derivation in terms of a Bayesian decision rule do not appear elsewhere according to our best knowledge. Furthermore, recognizing that we can relax the objective and optimize it in the space of policies is also a contribution. In the final version of the paper, we will carefully restate our contributions to make this clearer.

We had an omission of an assumption in the derivation of the objective. The first inequality in L587 only holds for large $T \rightarrow \infty$. This points to a similarity to Fiez et al. [1] being optimal in the asymptotic regime only as we explained in the Appendix. A better utility would require calculating the expectation of $ \mathrm{E}_{f \sim \text{GP}}[1/a_z^2]$ (see L574) which is unfortunately intractable. Were it possible to evaluate it would most certainly lead to a better utility. However, the gains are only to be seen for a very small regime of $T$ and quickly diminish. Based on this, the Eq. (3) in the paper should read:

$$
\min_{X_{\text{new}}}E_{f}\left[\sup_{z \in Z \setminus \{x_f^*\} }\log P(\mu_T(z) - \mu_T(x_f^*) \geq 0 | f)\right] \stackrel{\approx}{\leq}  \min_{X_{\text{new}}} E_f\left[ \sup_{z \in Z \setminus \{x_f^*\}} \frac{k_{X_{t} \cup X_{new}}(z,x_f^*)}{(f(z) - f(x_f^*))^2}\right]
$$

where we added the $\stackrel{\approx}{\leq}$ to denote that this is an approximate bound that will hold for large $T$, i.e., asymptotically. Note that the bound is on the log probability (this was a typo previously). The corresponding utility, the rest of the paper, and all empirical results are unchanged.

**Fixing the hyper-parameters for the experiments:** Good performance of Bayesian optimization is strongly dependent on such hyper-parameters, therefore there is a need to use previous data or expert knowledge to either fix the hyper-parameters (we use empirical Bayes, by giving all algorithms the same hyper-parameters learnt via marginal-likelihood on a small random hold-out set) or put a good prior while estimating them as we go (full Bayes). We fix them to compare without GP hyper-parameter choice being a co-founding effect since we are interested in quantifying the effect of the algorithm improvement. Note that all competing algorithms benefit or suffer from this. Investigating the best way of learning the GP's parameters would be interesting future work.

**References**

[1] Fiez, Tanner, et al. "Sequential experimental design for transductive linear bandits." Advances in Neural Information Processing Systems 32 (2019).

---

### Decision · Program_Chairs · 2024-09-25

**Decision:**

Accept (poster)

**Comment:**

This paper presents an approach to incorporating transition constraints through Markovian state dynamics into a Bayesian Optimization framework. Reviewers generally agree that the problem formulation is novel, well-motivated, and well-illustrated by the empirical results. Reviewers also found the solution approach to be principled and novel. However, as it (necessarily, due to the problem complexity) relies on a number of approximations and relaxations, I found keeping track of the various components somewhat challenging and therefore ask the authors to consider ways of helping the reader with this. In addition, reviewers raised some valid points about discussing additional related work that the authors should take into account. Finally, while there were some questions about the technical details of the algorithm that were not fully resolved in the rebuttal discussion, I believe that those do not invalidate the results and thus the value of the paper.